# Selective feeding in Southern Ocean key grazers—diet composition of krill and salps

Nora-Charlotte Pauli [1,2✉], Katja Metfies[2,3], Evgeny A. Pakhomov[4,5,6], Stefan Neuhaus[2], Martin Graeve [2], Philipp Wenta [1], Clara M. Flintrop [2,7], Thomas H. Badewien[1], Morten H. Iversen [2,7✉] & Bettina Meyer [1,2,3✉]

Over the past decades, two key grazers in the Southern Ocean (SO), krill and salps, have experienced drastic changes in their distribution and abundance, leading to increasing overlap of their habitats. Both species occupy different ecological niches and long-term shifts in their distributions are expected to have cascading effects on the SO ecosystem. However, studies directly comparing krill and salps are lacking. Here, we provide a direct comparison of the diet and fecal pellet composition of krill and salps using 18S metabarcoding and fatty acid markers. Neither species' diet reflected the composition of the plankton community, suggesting that in contrast to the accepted paradigm, not only krill but also salps are selective feeders. Moreover, we found that krill and salps had broadly similar diets, potentially enhancing the competition between both species. This could be augmented by salps' ability to rapidly reproduce in favorable conditions, posing further risks to krill populations.

[1] Institute for Chemistry and Biology of the Marine Environment, University of Oldenburg, Oldenburg, Germany. [2] Alfred Wegener Institute, Helmholtz Centre for Polar and Marine Research, Bremerhaven, Germany. [3] Helmholtz Institute for Functional Marine Biodiversity (HIFMB), Oldenburg, Germany. [4] Institute for Oceans and Fisheries, University of British Columbia, Vancouver, BC, Canada. [5] Department of Earth, Ocean, and Atmospheric Sciences, University of British Columbia, Vancouver, BC, Canada. [6] Hakai Institute, Heriot Bay, BC, Canada. [7] MARUM and University of Bremen, Bremen, Germany. ✉email: nora-charlotte.pauli@awi.de; morten.iversen@awi.de; bettina.meyer@awi.de

Antarctic krill (*Euphausia superba* Dana, 1850; hereafter krill) and the tunicate *Salpa thompsoni* Foxton, 1961 (hereafter salp) are among the dominant grazers in the Southern Ocean (SO)[1,2]. Krill is a key species in the SO ecosystem and an important link between phytoplankton and higher trophic levels such as whales, seals, and penguins[3,4], while salps are a less important food source for these apex predators[5]. Krill and salps can re-package large amounts of the primary production into large, carbon-rich and fast-settling fecal pellets, and thus play an important role in biogeochemical cycles and carbon export in the SO[1,6–8].

Krill are distributed throughout the entire SO[9], however, over 50% of their biomass is located in the southwest sector of the Atlantic Ocean (20–80°W), with particularly high densities in the western Antarctic Peninsula (WAP) region[9,10]. The WAP is currently one of the fastest warming regions worldwide[11], where increases in mean winter air temperature by 5–6 °C since 1950[12] have resulted in a reduced duration of sea ice coverage and a 10% decline in the extent of winter sea ice per decade[13,14]. Krill biomass is positively correlated to sea ice and cold temperatures[15,16], making them particularly susceptible to the observed climatic changes[17]. From 1926 to 2016, a southward shift of krill and declining abundances north of 60°S have been observed[10]. However, it remains debated whether there is a large-scale decline in krill abundances in the SO[10,15,18].

Opposite to krill, salp abundances are negatively correlated to sea ice cover[15,16]. Consequently, in response to climatic changes, an increase in the abundance of salps was observed and their southern distribution limit has shifted from 60°S to 65°S since 1980[1,15,16]. This has resulted in a greater overlap between the ranges of salps and krill, making them direct competitors for food and habitat in the WAP region[1,16,19]. Salps have already been documented as having replaced krill as the dominant grazer on small spatial scales during salp blooms at the northern WAP[20]. A long-term shift from krill to salps is expected to trigger a cascade of short- and long-term changes in the pelagic ecosystem of the western Atlantic sector of the SO[1,13] as the two organisms occupy very different ecological and spatial niches[1,20]. Krill and salps differ remarkably in their life cycles, reproduction, the way they fuel the lower food web by organic matter release and the upper tropic levels as prey, as well as in their importance for the SO fishery[16,20,21]. However, studies directly comparing the ecological role of krill and salps, particularly with respect to their diet and role in the carbon cycle, are lacking.

Both krill and salps are filter feeders, but differ in their feeding modes, potential prey-size spectrum and diet composition (Table 1). Krill are selective feeders with a diatom-dominated diet and were shown to prefer diatoms over smaller prymnesiophytes and cryptophytes in incubation experiments[22–24]. In contrast, the diet of salps mainly reflects the composition of the available plankton community, thus salps are assumed to be non-selective, indiscriminate feeders[1,25]. Selective feeding, i.e. the selection of particular prey items while avoiding or rejecting others, is displayed by most zooplankton[26]. The process of prey selection may be defined mechanically for example by the mesh size of the filtering apparatus ('passive selection'), or can be based on chemical cues or mechano-receptors ('active selection'), e.g. in krill[26–28]. In the pelagic environment with unevenly distributed food quantity and varying quality, prey selection is an important factor to balance the energetic costs of foraging against food quality and quantity and to adjust to changing conditions[27,29].

Selective feeding mechanisms can exert strong control on nutrient turnover, primary production and biogeochemical processes[27]. Consequently, the different feeding modes of krill and salps might have consequences for the plankton and grazer community structure at the WAP. Salps might outcompete krill for food during non-bloom periods, negatively affecting krill reproduction[30]. This effect would be compounded by salps ability to promptly respond to favorable conditions through asexual reproduction, allowing them to reproduce rapidly and form large swarms[5,25]. Additionally, over the past decades the plankton community composition in the northern part of the WAP has shifted from larger diatoms species to smaller flagellates[13]. This shift may favor salps over krill, as krill are less effective than salps at feeding on small cell sizes[31]. Recent studies found indications for selective feeding of doliolids, a group closely related to salps, in the Atlantic Ocean[32], and suggested that the diet of Mediterranean salps is determined by prey taxonomy, rather than size[33]. Yet, there are no recent studies on the diet of salps in the Southern Ocean and studies that simultaneously compare the diet composition of krill and salps from the same region are lacking. However, this knowledge is essential to understand the impact of a shift in dominance from krill to salps. The implications of such a shift based on earlier studies are difficult to assess, because each species was examined separately and their diets were studied using different methods (Table 1).

In recent years, advances in next-generation sequencing and metabarcoding (i.e. the amplification of a standardized region of

**Table 1 Feeding modes and diet composition of krill and salps.**

|  | *Euphausia superba* | *Salpa thompsoni* |
|---|---|---|
| Feeding mode | • Feeding basket formed by thoracic legs, filtering of particles through a fine net of setae[44,93]<br>• Feeding independent from active swimming, feeding rates can be adjusted[24,44] | • Mucous net deployed in the pharyngeal cavity retaining particles from water pumped from the anterior to the posterior opening[45,94]<br>• Feeding and locomotion are continuous processes[45], no adjustment of feeding rates[1,94] |
| Potential prey size range | • 2–3 µm to ~1 mm[57,93] | • Submicron <1 µm to >1 mm[25,45,64] |
| Diet composition summary | • Diatom-dominated diet, mainly herbivorous[24,44] | • Food generalists, diet reflects available plankton community[25,45] |
| Microscopy/visual inspection | • Diatom-dominated, autotrophic flagellates, dinoflagellates, tintinnids[37,59] | • Diatom-dominated, radiolarians, silicoflagellates, dinoflagellates[38,95] |
| Fatty acids | • Diatoms, copepods, foraminifera, flagellates, athecate dinoflagellates[57,59,96] | • Flagellates, moderate diatom contribution, copepods[38] |
| Metabarcoding | • Diatom-dominated, cercozoans and copepods[35] | • Dinoflagellate-dominated, few diatoms[34] |
| Other DNA-based methods | • Diatom-dominated diet incl. silicoflagellates, copepods, cercozoa dinoflagellates, ciliates, cercozoans[37,97] | • NA |

Overview of the available literature data on the feeding mode and diet composition of Antarctic krill (*Euphausia superba*) and salps (*Salpa thompsoni*). Studies on the diet composition of krill and salps are grouped by the method used; prey items are listed by abundance as reported in the cited literature.

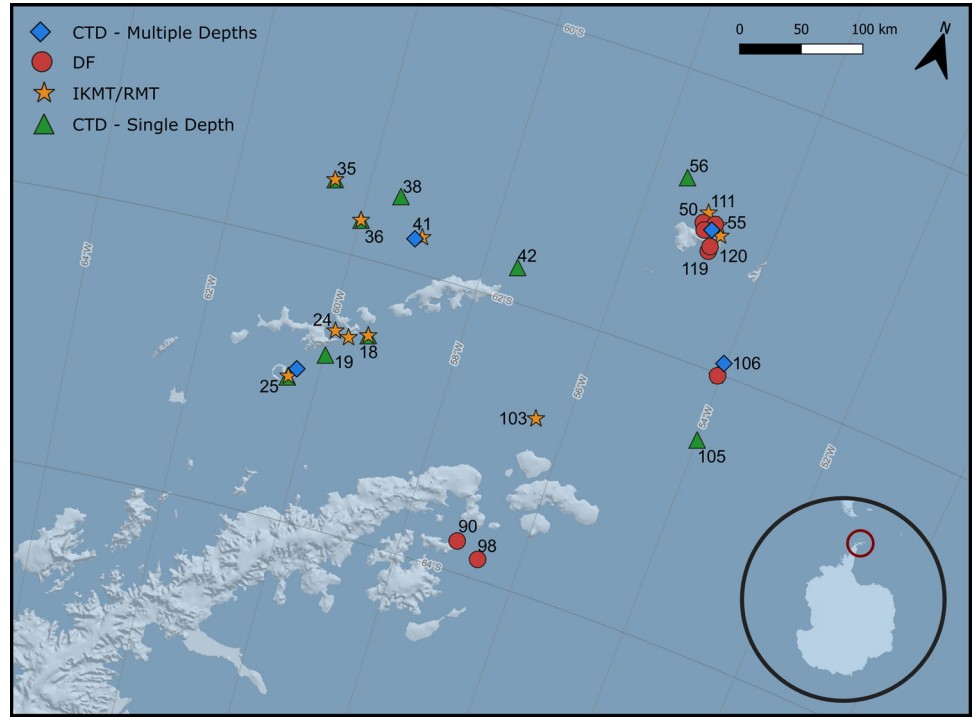

**Fig. 1 Sampling stations and deployed devices at the Antarctic Peninsula.** All sampling stations are shown (cf. Supplementary Table 3) along with symbols depicting the different sampling devices: Blue diamonds depict water samples collected with a CTD rosette at multiple depths, green triangles depict water samples collected with a CTD rosette at a single depth in the chlorophyll maximum layer. Yellow stars depict krill and salps collected by tows using IKMT or RMTs. Red circles depict drifting sediment traps, from which fecal pellets were collected. The map was created using the software Quantartica[98].

the DNA and subsequent sequencing and comparison to a reference database) has enhanced the possibilities to detect a broader prey size range including rare species and have already been applied for krill and salps[34–36]. Moreover, the combination of molecular and traditional methods such as microscopy and fatty acid markers has allowed for a comprehensive insight into short and long-term dietary habits[34,37,38].

Here, we investigate the diet composition of krill and salps by comparing diet analyses and fecal pellet content with the ambient plankton community in the WAP region. We used a combined analytical approach of 18 S rRNA metabarcoding and fatty acid analyses to reveal long- and short-term diets of krill and salps and to identify feeding preferences of both species. Moreover, we evaluate their roles in biogeochemical cycling by linking their ingestion to the composition of the egested fecal pellets. This will provide valuable information to predict how a shift in phytoplankton composition will affect the distribution of krill and salps, which in turn influences the efficiency of the biological carbon pump in the WAP region.

## Results

**Composition of stomach content, fecal pellets, and the ambient plankton community.** The diets of krill and salps were compared by sequencing genomic DNA extracted from the stomach contents and fecal pellets of organisms collected from the same regions along the AP (Fig. 1). To identify selective feeding behavior, we compared the stomach contents of krill (61 samples across 10 stations) and salps (60 samples across 10 stations) to the ambient plankton community (10 samples across 10 stations of 2 L each). In addition, the composition of fecal pellets (FP) produced by krill and salps were compared ($n = 14$ for krill FP, $n = 11$ for salp FP). The raw sequencing data were refined (see material and methods), and the final, analyzed dataset consisted

of 1765 unique amplicon sequence variants (ASV) in 156 samples across five sampling groups: krill stomach content, krill FP, salp stomach content, salp FP, and the ambient plankton community.

A principal component analysis (PCA) of the refined dataset revealed three main clusters, which were associated with all five sampling groups (Fig. 2), accounting for 43.6% of explained variation in the first two dimensions (36.6%, and 7%, respectively), while the remaining dimensions explained less than 5% each. An analysis of similarity (ANOSIM) revealed significant differences between all sampling groups ($p = 0.01$, test statistic $R = 0.46$, permutations = 999); however, the clusters of stomach content and fecal pellets of both species each largely overlapped (Fig. 2).

The 18S metabarcoding libraries of the water column were dominated by dinoflagellates, mainly represented by the genus *Gyrodinium* (Dinophyceae, 39.5%; Fig. 3a). In addition, diatoms (Bacillariophyta, 19.4%), and Prymnesiophyceae (mainly *Phaeocystis* sp., 10.9%) represented the most abundant taxa. Copepods, including the genera *Oithona* and *Metridia*, accounted for 6.3%, and the parasitic dinoflagellate group Syndiniales for about 5% of all sequences in the plankton community. Less abundant taxa included Spirotrichea (ciliates), Picozoa (small, unicellular heterotrophs), and Cryptophyta (unicellular flagellates), among others. About 8% of the plankton community comprised taxa with a relative abundance of less than 1.5%, including small flagellates (Filosa-Thecofilosea; Supplementary Table 1).

Overall, the sequence libraries obtained from the stomach content of krill and salps did not reflect the dinoflagellate-dominated plankton community nor the high proportion of diatoms observed in the water samples. The 18S metabarcoding libraries of the krill stomach content included a high share of crustacean sequences (26.5%; Fig. 3c), which were mainly assigned to the copepod genera *Calanus* and *Oithona*. The

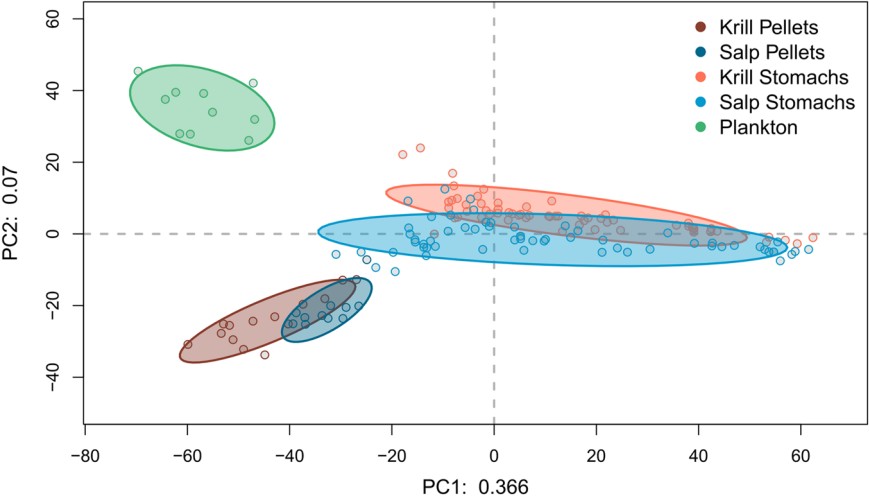

**Fig. 2 Principal component analysis (PCA) of the 18S metabarcoding libraries.** The PCA shows the refined, centered-log-ratio transformed dataset containing 156 samples with each dot representing an individual sample. Ellipses are drawn around the five pre-defined sampling groups krill fecal pellets (dark red), salp fecal pellets (dark blue), krill stomach content (light red), salp stomach content (light blue), and the ambient plankton community (green) with a confidence interval of 75%. The explained variation in decimals for the first two principal components (PC) is shown on the x- and y-axis, respectively.

amount of dinoflagellates (Dinophyceae) and diatoms was markedly reduced compared to the plankton community (14.5 and 12.2%, respectively), while the parasitic Syndiniales group was enriched in krill stomachs accounting for 12.4% of the sequences. In addition, salps constituted almost 6% to the stomach content of krill.

The sequence libraries obtained from the stomach content of salps were dominated by the parasitic dinoflagellate group Syndiniales (31%; Fig. 3b) and contained a high share of small flagellates (*Ebria* sp., Filosa-Thecofilosea; 20.7%). Other dino-flagellates (Dinophyceae) and diatoms were less abundant compared to the plankton community accounting for 18.1%, and 7.4% of the sequences, respectively. Crustacean remains in salp stomachs contributed to 13.1% of all sequences and were mainly assigned to copepods (*Oithona*, *Metridia*, *Calanus*), but also included some unidentified Malacostraca sequences. In contrast to krill, salp stomach samples showed some regional patterns (Supplementary Fig. 1).

The composition and relative abundance of taxa found in the 18S metabarcoding libraries of the FP of both species differed from the stomach content and from the ambient plankton community. In krill FP, small flagellates (*Ebria* sp., Filosa-Thecofilosea) accounted for the vast majority of sequences (56.4%; Fig. 3e). The share of dinoflagellates (Dinophyceae) was almost constant in krill FP compared to their stomach content, accounting for 13.9% of the sequences. The amount of crustaceans, parasitic dinoflagellates, and diatoms was reduced in krill FP compared to their stomach content, and prymnesio-phytes and salp sequences were almost absent. Salp FP were dominated by diatoms (Bacillariophyta, 36.2%; Fig. 3d) and small flagellates (Filosa-Thecofilosea, 33%). Both groups were notice-ably enriched in comparison to the stomach content of salps, while the share of dinoflagellates and crustaceans, and especially that of the parasitic Syndiniales group was reduced.

A repeated, quantitative PCA supported the general patterns of the relative abundances, including only ASVs with significant differences within or between any of the groups (effect size >1) or those, that were compositionally associated (ρ > 0.5; Supplemen-tary Fig. 2). This showed that dinoflagellates (Dinophyceae) were more abundant in plankton samples, while small flagellates (Filosa-Thecofilosea) were more abundant in the fecal pellets of

krill and salps. In addition, unsupervised clustering using Euclidean distances supported the observed separation of plankton, stomach content, and FP clusters (Supplementary Figure 3). Mapping other metadata variables on the PCA biplot revealed no apparent clusters by region (Supplementary Fig. 4), indicating that the separation by sampling group (stomach content, fecal pellets, plankton) adequately explained the structure of the data.

**Feeding selectivity**. Across all stations, salps showed a strong preference for small flagellates (Filosa-Thecofilosea, Ivlev's selectivity index = +0.88; Fig. 4), Syndiniales (+0.73), and Chrysophyceae (+0.78), while Prymnesiophytes (−0.90), Pelago-phyceae (−0.87), Picozoa (−0.90), and Cryptophyceae (−0.96) were avoided. Krill selectively fed on copepods (+0.61), poly-chaetes (+0.58), and golden algae (Chrysophyceae, +0.89; Fig. 4), while avoiding radiolarian protozoans (Acantharia, −0.85) and Cryptophyceae (−0.71).

In addition to the plankton community sampled at a single depth in the chlorophyll maximum layer, at four of the ten stations water samples were collected at the surface, 100 m, and 200 m depth (Supplementary Figure 5). The preference of salps for Filosa-Thecofilosea was also observed across the different depths (>0.4; Supplementary Figure 6) and a preference for Syndiniales was observed at all stations except for one (St. 25). Similarly, the avoidance of prymnesiophytes and the other avoided groups was apparent across all depths (−0.47 to −1). For krill, the preference for copepods was apparent across all depths and the preference for polychaetes was observed at all stations except for one (St. 25).

**Fatty acid composition**. In addition to the short-term diet stu-died from the stomach content, we used long-term dietary mar-kers by extracting fatty acids from tissue samples of krill (n = 21) and salps (n = 22). Overall, we identified 30 different fatty acids, 15 of which had a share of more than 1.5% of total fatty acids (Supplementary Fig. 7, Supplementary Table 2, see Supplemen-tary Results for details on lipid classes and fatty alcohols). Diatom marker fatty acids (16:1($n$−7), 20:5($n$−3)) differed significantly between both species and were higher in krill (median = 30.6% of total fatty acids, p = 0.002; ANOVA; Fig. 5, Table 2) compared to

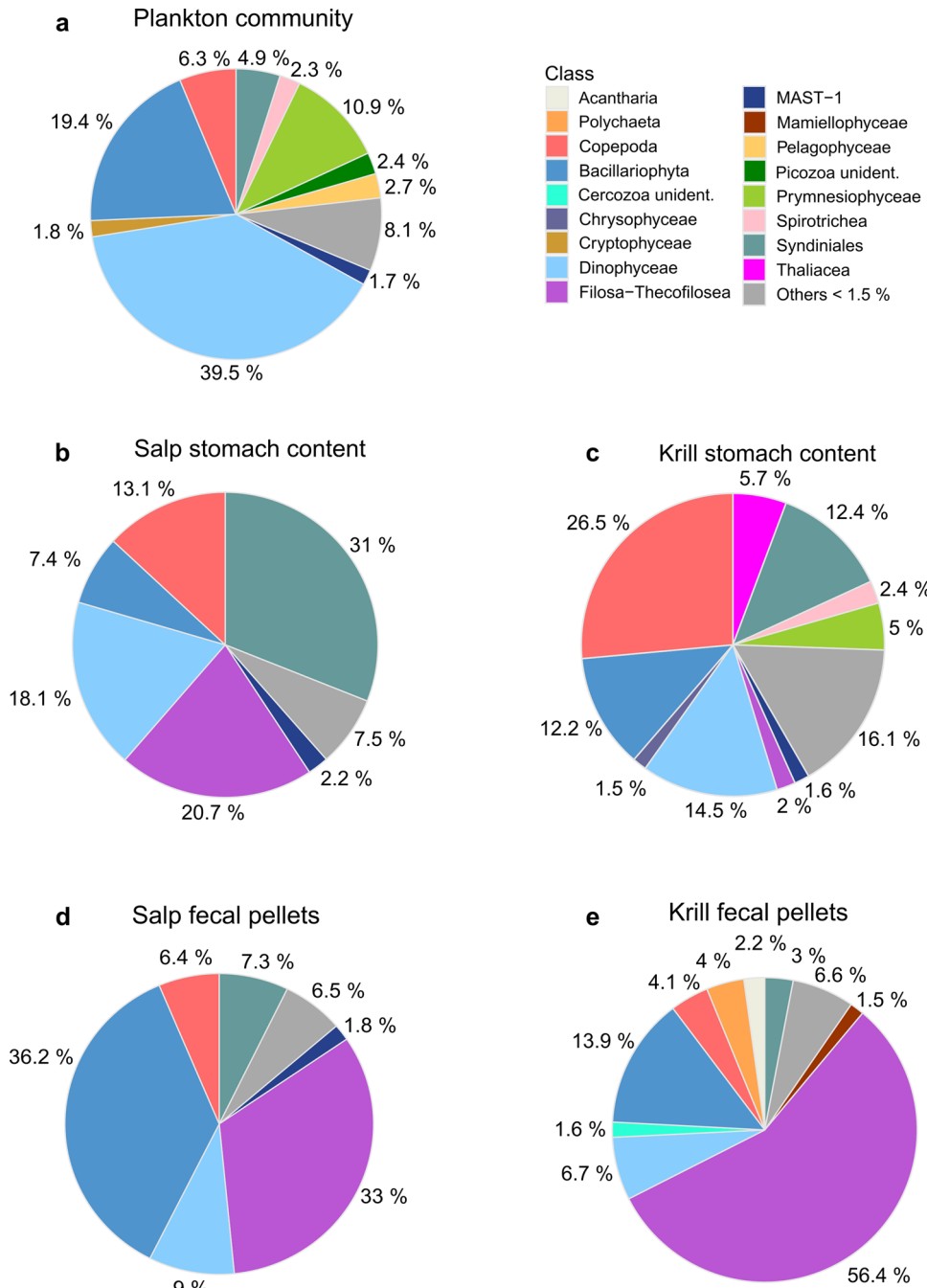

**Fig. 3 Composition of the 18S sequencing libraries for all sampling groups.** Relative abundances are shown in percent of all taxa aggregated on the taxonomic level of 'Class' for each of the sampling groups: (**a**) ambient plankton community ($n = 10$), (**b**) salp stomach content ($n = 60$), (**c**) krill stomach content ($n = 61$), (**d**) salp fecal pellets ($n = 11$), and (**e**) krill fecal pellets ($n = 14$). Relative abundances were calculated as the mean over all samples within each of the five groups. Taxa with a relative abundance of <1.5% within each group were pooled and are shown as 'Others'.

salps (median = 17.1%). There was no significant difference between the regions, nor an interaction between species and region. The sum of the fatty acids 16:1($n-7$), 16:2($n-4$), 16:3($n-4$), and 16:4($n-1$) was also a marker for diatoms and differed significantly between species ($p = 0.001$, $F = 12.37$, Df = 1; ANOVA) with higher values in krill compared to salps. To study the effect of sex and length on marker fatty acids within each species, we performed separate tests for krill and salps. For diatom marker fatty acids, this revealed no significant effects for either species. Dinoflagellate marker fatty acids (18:4($n-3$), 22:6($n-3$)) showed no difference between species (median = 14% for each species, $p = 0.665$, ANOVA, Fig. 5, Table 2) nor between

regions. However, when testing each species separately, there was a significant effect of sex as well as a significant interaction of sex and length for krill ($p = 0.006$, $F = 7.13$, Df = 2, and $p = 0.046$, $F = 4.68$, Df = 1, respectively; ANOVA), and a significant effect of sex in salps ($p = 0.01$, $F = 8.39$, Df = 1; ANOVA). Pairwise comparisons using a Wilcoxon rank-sum test revealed that these significant effects reflected differences between female and male krill (p = 0.037), and between aggregate and solitary salps (p = 0.019), respectively. Marker fatty acids for calanoid copepods (20:1($n-11/n-9/n-7$), 22:1($n-11/n-9/n-7$)) were significantly higher in salp tissue compared to krill (median 0.05% vs. 0.02%, $p < 0.001$; Kruskal–Wallis rank sum test; Fig. 5,

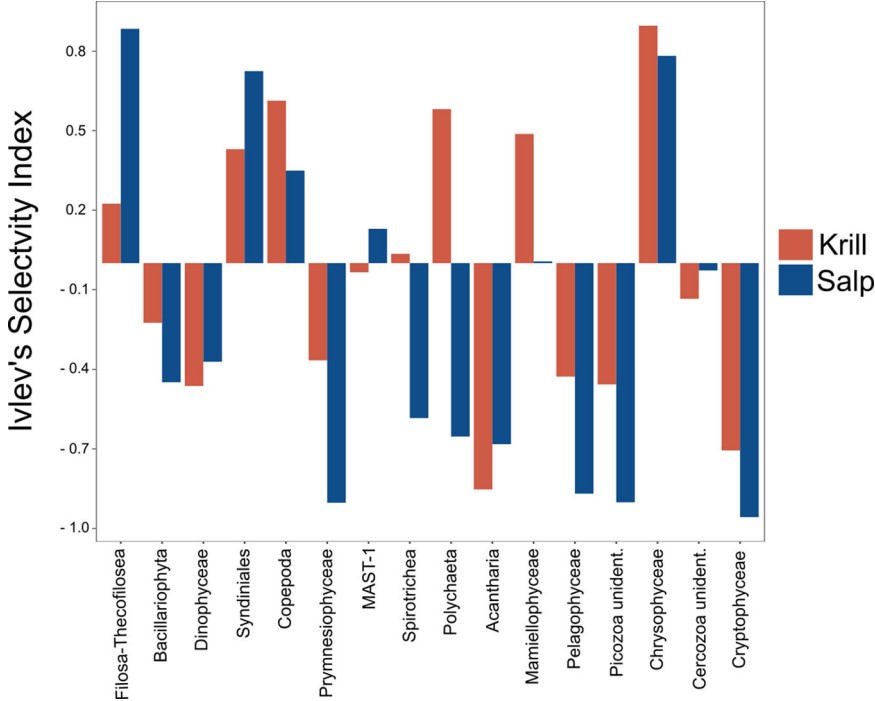

**Fig. 4 Feeding selectivity of krill and salps.** Ivlev's selectivity index was calculated based on the relative abundance of the 18S sequencing libraries of the plankton community sampled from the chlorophyll maximum layer ($n = 10$) and the stomach contents of krill and salps ($n = 61$ and $60$, respectively). A selectivity index of +1 indicates preference of a prey taxon, while an index of –1 indicates avoidance. Selectivity of krill is depicted by red bars, salps selectivity by blue bars.

Table 2). There was no significant effect of sex or length for both species.

Multidimensional scaling using a log-ratio analysis showed a clear clustering of both species along the first dimension, which explained 25.9% of variation (Fig. 6). The second dimension explained 10.5% of the variation and was mainly defined by fatty acids $16:1(n-5)$, $18:1(n-5)$, and $18:4(n-3)$ on the one, and $20:5(n-3)$ on the other side. One krill sample from the South Shetland Islands was an outlier, as it clustered closer to the salp samples than to the rest of the krill samples. This krill sample was particularly poor in lipids (<1% of dry weight), while the mean share of lipids per dry weight in krill tissue was 6.7% compared to 0.97% in salps. Fatty alcohols were generally low compared to fatty acids, with slightly elevated values of $14:0$ and $18:1(n-9)$ in single salp tissue samples.

## Discussion

In this study, we investigated the diet composition and feeding preferences of krill and salps in several regions along the western Antarctic Peninsula (WAP) and compared their diet to the ambient plankton community and the content of their fecal pellets (FP). The plankton community was dominated by dinoflagellates, diatoms and prymnesiophytes, which is typical for an autumn (March–May) plankton community around the WAP[39]. Dinoflagellates were dominated by Gymnodiniaceae, coinciding with previous studies during this season[40]. Prymnesiophytes were mainly represented by *Phaeocystis* sp., which often dominate ice algal blooms at the WAP[41]. The main fatty acid composition in the tissue of krill and salps was consistent with previous studies on the phytoplankton fatty acids in the SO, indicating a mixed plankton community comprising diatom ($16:1(n-7)$, $20:5(n-3)$) and non-diatom markers ($22:6(n-3)$ and $18:4(n-3)$)[42,43].

Comparisons of the ambient plankton community composition to the stomach and fecal pellet contents of krill and salps revealed the two species' diets did not reflect the composition of the

ambient plankton community, suggesting selective feeding of both species. Selective feeding has previously been observed for krill[24,44], as they were shown to prefer diatoms over prymnesiophytes in laboratory experiments[23]. In addition, the broad range of feeding behavior observed in krill, including feeding in the water column, under sea ice, and on the sea floor, strongly suggests that krill are capable of adjusting their feeding strategy depending on the available food source[24]. In contrast, salps were assumed to be continuous and non-selective filter feeders[1,25], with diets generally reflecting the composition of the available plankton community[45]. However, the salp stomach content in this study was significantly different from the ambient plankton community, and salps selectively fed on Filosa-Thecofilosea (small flagellates) and golden algae over prymnesiophytes and other unicellular algae. The feeding selectivity of krill and salps was confirmed across different depths from the surface to 200 m at several stations. Furthermore, there was no significant difference between krill and salp diet composition, suggesting that contrary to previous hypotheses salps might be as selective in their feeding behavior as krill.

It remains unclear how salps perform such selective feeding behavior. On-board video observations obtained during this study showed that salps are agile swimmers, often performing back-flushes to empty their feeding tract. This back-flushing was performed by a jet propulsion out of the frontal opening (anterior aperture) causing the salps to move backwards, whereupon they would change their swimming direction (Supplementary Movie 1). This behavior may allow for selective feeding by rejecting non-desirable food via back-flushing and subsequent swimming to other feeding grounds.

In this study, the diet of salps was dominated by (dino-) flagellates, which is in accordance with previous results from the Lazarev Sea[34]. Moreover, fatty acid analyses have indicated that salps have a flagellate-based diet year-round with a moderate amount of diatoms, and the overall fatty acid composition of salps

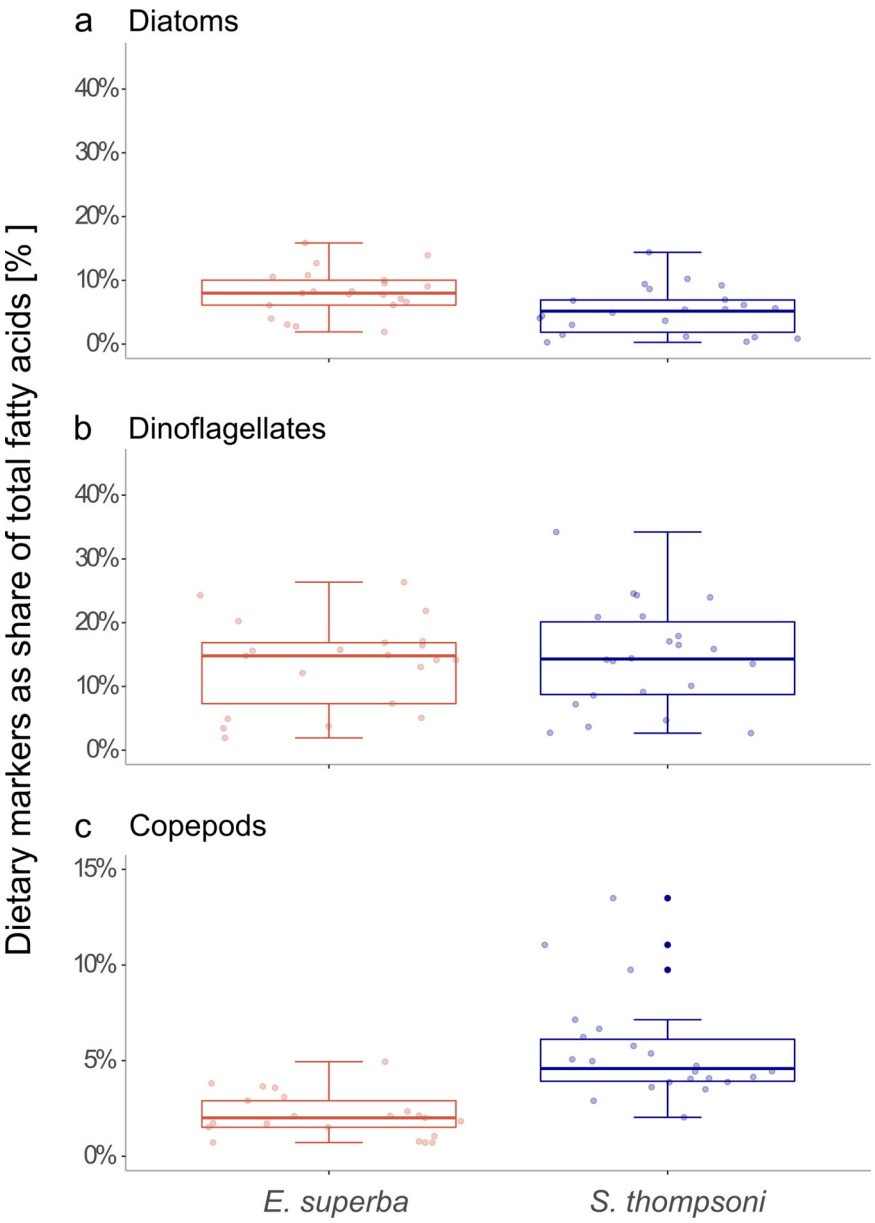

**Fig. 5 Marker fatty acids for major plankton groups.** Fatty acids were identified from the tissue of krill (*n* = 21) and salps (*n* = 22). The sum of marker fatty acids for three plankton groups: (**a**) diatoms, (**b**) dinoflagellates, and (**c**) calanoid copepods as percentage of total fatty acids is shown on the y-axis for krill (red) and salps (blue), respectively. Dots represent individual samples.

agrees with previous studies in the WAP region[38,46]. Besides flagellates, crustaceans accounted for about 13% of the diet of salps, including copepods, as well as unidentified Malacostraca sequences, which were blasted and identified to be krill sequences. Salps have previously been suggested to feed on krill eggs and larvae[47], and krill remains have been found in salp guts[48].

Despite the similarities between the stomach contents of krill and salps, we observed distinct differences when comparing the composition of their FP. This indicates differences in digestion and assimilation efficiency of the food ingested by the two species. The relative abundance of diatoms was significantly higher in salp FP (36.2%) compared to their stomachs (7.4%), supporting the notion that salps cannot digest diatoms efficiently due to the lack of mandibles or morphological structures such as a gastric mill needed to break down silica frustules[38,49]. However, we found 17% of diatom fatty acid markers in salp tissue, indicating the assimilation of diatoms. This is consistent with previous fatty acid

studies showing that salps are able to partly digest diatoms[50]. During visual inspections of some salp stomachs on board, we observed krill FP, suggesting that salps ingested those krill FP. Thus, the ingestion of already broken diatom cells inside of krill FP may allow salps to digest and assimilate diatoms. Overall, the composition of salp FP was more similar to their stomach content than krill FP to krill's diet, suggesting that krill more efficiently digest their prey than salps.

Further differences between the stomach content and FP composition were found in unicellular flagellates (*Ebria* sp., and Cryomonadida), which only contributed 1–3% to the plankton community and to the diet of krill, but were considerably enriched in FP, especially in those produced by krill (56.4%). Ebridian flagellates can reach high densities within plankton communities and have an internal solid siliceous skeleton[51], which might cause limited digestion by their predators. Another potential reason for such high relative abundances in the excreted material might be

that the flagellates colonized FP in the water column after they were produced. Dinoflagellates (Dinophyceae) and ciliates (Spirotrichea) showed a trend towards higher abundances in FP from 100 m compared to 300 m, indicating that those taxa graze on fecal pellets in the upper water column. Krill and salp FP both play a major role in the carbon cycle due to their high sinking velocities[52,53]. Earlier studies proposed that the diet of krill and salps affects the sinking velocities and consequently the export of FP[54]. Due to the ballasting effect of minerals such as opal, pellets produced on a coccolithophore or diatom-based diet have higher sinking velocities than pellets produced on a flagellate diet[55]. Here, salp FP contained a considerably higher share of diatoms than krill pellets (36.2% vs. 13.9%), suggesting that salp FP could sink faster than those produced by krill, coinciding with previous observations[8,56].

Crustaceans accounted for about one third of the prey sequences in krill stomachs, and were mainly represented by the copepod genera *Calanus*, *Oithona*, and *Metridia*, agreeing with previous stomach content studies on krill[24]. This suggests that we

observed a feeding strategy common for krill in autumn and winter when phytoplankton is scarce used to supplement their otherwise mainly herbivorous diet[24]. Our sequencing results were further supported by long chain fatty acids (20:1, 22:1) indicative of calanoid copepods in the tissue of krill[57,58]. A possible contribution of euphausiid molts in the diet of krill which also serve as a supplementary energy source during autumn and winter[59] might be masked in this study, as for methodological reasons all sequences attributed to the group of euphausiids and close relatives were excluded from the analyses of krill samples because they were considered predator DNA. Overall, the amount of total lipids per dry weight determined for krill in this study was considerably low (6.7%) compared to previous studies conducted in autumn (28.2%) when lipid reserves typically accumulate for winter[60]. However, while those previous studies used whole animals including the digestive tract, we analyzed tissue and stomach samples separately to be able to distinguish between short- and long-term dietary markers and to distinguish assimilation from ingestion.

Previous experiments have shown that krill are capable of gaining up to 9% of their body carbon per day by feeding on salps[1,61]; however, the true dynamics of krill feeding on salps remain unclear. Studies investigating this in situ and in regions where both species co-occur are so far lacking. In this study we were able to directly compare the diets of krill and salps sampled from the same region at the same time and show that the diet of krill contained about 6% salp sequences, suggesting that krill were either directly feeding on salps or on their remains. During on-board video recordings, we observed an individual krill holding on to a small chain of salps with its feeding basket and swimming with it for more than one minute before letting go (Supplementary Movie 2). Although we were not able to observe the krill feeding directly on the salps, the attachment of the krill to the salp chain support previous observations of krill preying on salps[61]. Our 18S metabarcoding results provide empirical evidence for feeding on salps by krill and highlight the importance of applying molecular techniques to resolve diet patterns, as the gelatinous body of salps is more rapidly dissolved in the digestive tracts of predators than other hard-bodied prey items and may therefore be underrepresented by traditional,

**Table 2 Statistical results of fatty acid markers between krill and salps.**

| Response variable | Explanatory variable | Df | *F*-value | *P*-value |
|---|---|---|---|---|
| Diatom marker | Species | 1 | 11.103 | **0.002** |
| | Region | 5 | 1.613 | 0.185 |
| | Species*Region | 4 | 0.923 | 0.463 |
| Dinoflagellate marker | Species | 1 | 0.196 | 0.661 |
| | Region | 5 | 0.774 | 0.576 |
| | Species*Region | 4 | 0.425 | 0.790 |
| Copepod marker | Species | 1 | 26.613 | **<0.001** |
| | Region | 5 | 4.294 | 0.508 |

Results of the analysis of variance (ANOVA) testing for the effect of species and region on diatom and dinoflagellate marker fatty acids and of the Kruskal–Wallis rank-sum test to test for the effect of species and region on copepod marker fatty acids. Significant results are highlighted in bold. Degrees of freedom (Df), the test statistic (F-value for ANOVA, Chi-squared for Kruskal test) and P-value are shown.

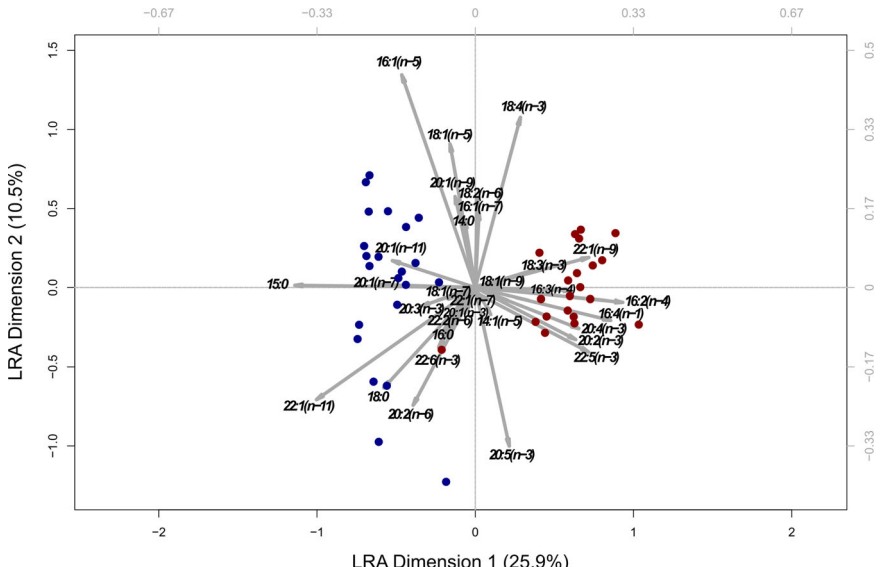

**Fig. 6 Results of the unweighted log-ratio analysis (LRA) of fatty acids profiles.** Fatty acids were extracted from the tissue of krill and salps ($n = 21$ and 22, respectively). The results of this analysis were plotted using a contribution biplot with a rescaling factor of three to allow for a better interpretation of dispersion. Krill samples are depicted in red and salp samples are shown in blue. Arrows depict the single fatty acids, while the length of the arrows corresponds to the relative contribution of the respective fatty acid to the explained variance. The explained variance for the first two dimensions is given in percent.

visual methods[5]. It was recently recognized that salps are of higher nutritional value than previously assumed and an increasing number of species is recognized to prey on salps[5]. The presence of salp sequences in krill stomachs might also indicate krill feeding on salp FP, which is of particular interest as salp FP are thought to play an important role in the carbon cycle. However, salp sequences accounted for only a small share (2.1%) in salp FP, suggesting that the larger share of salp sequences in the diet of krill derived from feeding on salps or salp remains.

The parasitic dinoflagellate group Syndiniales accounted for almost one third of the salp and 12.4% of the krill diet, while being less abundant in the plankton community (4.9%). Syndiniales are ubiquitous marine parasites (Alveolata), contributing up to 20–50% of 18S rRNA diversity in the Southern Ocean[62]. Compared to krill, salps might be able to graze more effectively on small dinospores (<10 μm), which are released by the parasites for reproduction[63,64]. However, the relative enrichment of Syndiniales sequences in the stomachs of either species versus the plankton community might indicate prior infections of salps or krill, which has previously been documented[65,66]. Another possible reason for the presence of Syndiniales in stomach and FP samples could be the ingestion of infected prey, such as copepods or other dinoflagellates[67].

Overall, we did not find significant differences in the diet composition of krill and salps. It is generally assumed that krill and salps feed on different prey size ranges and that krill are selective feeders while salps feed indiscriminately. Thus, it was assumed that their diets differ[31], while to date there are no studies that have directly compared their diet composition. Our study found krill and salps to have similar diets and supports previous research suggesting that the two species compete for food when they co-occur[1,30,68]. Though often considered to be only of minor importance, the direct competition for food might be one of several factors driving krill and salps apart spatially and temporally, especially when abundances of either species are high[1]. Moreover, in combination with the ability of salps to reproduce rapidly under favorable conditions, their selective feeding behavior and similar diet enable salps to outcompete krill. This poses an additional threat to krill populations, which are already affected by warming waters and are particularly susceptible to climate change[17]. In contrast to krill, due to their highly efficient feeding strategy and wide size spectrum, salps are also able to survive during times of low food availability and thrive rapidly when conditions improve, while krill depend on a longer juvenile development. Competitive removal of food resources may also be a crucial factor on a longer time-scale, especially as the habitats of krill and salps become increasingly similar and will overlap more due to global warming[1,16]. It has been suggested that if salps feed more efficiently than krill on small prey then they would benefit from a shift in the WAP phytoplankton community from diatoms to cryptophytes[31]. However, we observed selective feeding behavior of both krill and salps and did not find evidence for more efficient feeding on small prey by salps compared to krill. This indicates that the projected shift in plankton size along the WAP might not favor salps as previously suggested[31]. A shift in the grazer abundance, as it has already been observed in some parts along the WAP, might therefore not merely be associated to the available prey size range, but rather to the community composition itself. Future studies should further investigate the selective feeding behavior using both field and laboratory tests. If selective feeding behavior is a prevailing strategy for both species, this will likely have long-term effects on the distribution of krill and salps, enhancing the already observed increase in salp abundances and southward shift of krill, particularly in combination with the changing plankton community composition around the WAP.

## Methods

**Field sampling.** Samples were collected during the research cruise PS112 with RV Polarstern between March and May 2018 along the Antarctic Peninsula (AP). We sampled six different regions between 60° 44.53' S to 63°59.16' S and 53°55.39' W to 60°31.57' W (Antarctic Sound, Bransfield Strait East and West, Deception Island, Elephant Island, and the northern region of the South Shetland Islands). We compared the ambient plankton community with the stomach content and fatty acid composition of krill and salp tissue, as well as fecal pellet (FP) contents of both species in each of the sampling regions. If it was not possible to collect all samples from a single station, we chose samples from adjacent stations, resulting in a total of 26 stations across all study regions (Supplementary Table 3).

For stomach content and tissue analyses, krill and salps were collected at ten stations across all regions by oblique net tows with an IKMT (Isaacs-Kidd Midwater Trawl, mesh size 505 μm), or with a RMT (Rectangular Midwater Trawl, mesh size 320 μm) in the upper 200 m of the water column. The catch was onboard and inside within five minutes and the animals were measured, sexed, staged, frozen in liquid nitrogen, and stored individually at − 80 °C within approximately ten minutes. The stomachs of frozen krill and salps were dissected with a replicate quantity of ten individuals per species and station. The dissection was performed on ice to prevent thawing, the dissected stomachs and remaining parts of each individual were kept frozen until further analyses.

FP samples were collected at 15 stations across five of the six regions, while no samples were available from the South Shetland Islands. We collected FP produced by krill from free drifting sediment traps (hereafter drift traps), as well as freshly produced salp fecal pellets from incubation experiments. For this, freshly caught salps were placed into 20 L buckets filled with ambient, unfiltered seawater from the upper 5 m of the water column and kept in darkness for 6–12 h at in situ temperature (0–1 °C). Pellets were carefully collected from the buckets and either frozen in filtered seawater or filtered on a pre-combusted GF-F filter and stored at − 20 °C until further analyses. Ten drift trap deployments were conducted and each deployment lasted ~24 h to collect sinking materials from three depths: 100, 200, and 300 m. The drift trap array consisted of a surface buoy with an Iridium satellite sender providing the trap position, 14 smaller buoys acting as wave breakers, two glass floats for buoyancy, and four trap cylinders per collection depth (84.95 cm² collection area each). At each collection depth, the four trap cylinders were attached to a gimbal mount ensuring a vertical position of the collection cylinders during the deployment. After recovery of the traps, particles were allowed to settle for 12 h before the overlaying water was removed and the collected material was rinsed into a sample container with GF-filtered seawater and frozen at − 20 °C.

The ambient plankton community was analyzed using 18S metabarcoding of water samples collected using a CTD rosette (Sea Bird Scientific SBE911plus and Carousel Water Sampler SBE 32) equipped with 24 × 20 L Niskin bottles, with a sensor for chlorophyll measurements (FluoroWetlabECO AFL FL). At ten stations across all study regions, we collected a subsample of 2 L from the chlorophyll maximum layer (15–60 m), which was determined during the down cast. Water was filtered through a 0.4 μm membrane filter (Whatman Nucleopore, 47 mm polycarbonate) at 200 mbar using a peristaltic pump and filters were stored at − 80 °C until DNA extraction. Additionally, at four of the ten stations, water samples were collected for a vertical profile from the surface (3–14 m), chlorophyll maximum, 100 m and 200 m depth (Supplementary Table 3). These samples were consecutively filtered through three filter sizes, 10 μm, 3 μm, and 0.4 μm and stored at − 80 °C.

**DNA Isolation and amplification.** Genomic DNA was isolated from krill and salp stomachs, and from plankton samples using the NucleoSpin® Plant II Mini Kit, while DNA from fecal pellets produced by both species was extracted with the NucleoSpin® Soil Kit (both Macherey-Nagel, Germany). Samples were defrosted immediately prior to the DNA isolation, using a slightly modified manufacturer protocol, as described in the following. Salp stomach samples, which exceeded the 10 mg limit of the DNA kit, were divided into subsamples after mechanical homogenization in an appropriate ratio of lysis buffer. Elution was performed in two steps of each 30 μl of PE buffer for stomach and plankton samples, and one step of 30 μl for fecal pellets. The eluted DNA of the salp stomach subsamples was pooled and DNA of the water samples for the vertical profile were pooled using 5 μl per size fraction. Subsequently, DNA was quantified using the QuantiFluor® double stranded DNA system (Promega, USA). Genomic DNA isolates were then used to amplify a 436 bp 18S rRNA fragment of variable region V4 using 2.5 μl of DNA (5 ng μl⁻¹), 12.5 μl of KAPA HiFi HotStart ReadyMix PCR Kit (Kapa Biosystems), and 5 μl of each forward and reverse primer (1 μM each). Amplification was based on the primers 528iF 5'-GCGGTAATTCCAGCTCCAA-3' and 964 R 5'-ACTTTCGTTCTTGATYRR-3'[69]. The polymerase chain reaction (PCR) was conducted with 25 cycles of 95 °C for 30 s, 55 °C for 30 s and 72 °C for 30 s, followed by 72 °C for 5 min and subsequent cooling to 4 °C. Samples were prepared for sequencing following the 16S metagenomic sequencing library preparation protocol (Illumina®) with a final DNA concentration of 12 pM in the library pool with 20% PhiX control (14 pM for krill stomach samples). Two to three template controls without DNA were carried through all steps of the PCR to account for potential contamination. In addition, template controls were randomly included in the library preparation steps. We did not find evidence for contamination.

A common methodological issue when sequencing stomach or gut content samples is that predator DNA often swamps prey sequences[70]. Preliminary tests on

the stomach content samples of krill and salps revealed that this was the case for both species, but to a much higher degree for krill. Therefore, we designed a blocking probe (5′-GACGGGCTTTAGCGTTC-3′) binding to the krill rRNA gene between the applied primers using the ARB software tool[70,71] (see Supplementary Fig. 8 and Supplementary Material and Methods for details on the probe design). The blocking probe was added to the PCR mix mentioned above (19 μM, 5 μl). Amplification of DNA isolated from krill stomach samples with the blocking probe resulted in a lower total DNA yield. Therefore, we amplified and sequenced the stomach content samples of krill in triplicates to be able to pool the sequencing information from different reactions, as well as to account for the variability between samples caused by the expected low sequencing yield.

**Amplicon sequencing and sequence processing.** Amplicon sequencing was performed on an Illumina MiSeq sequencer producing 2×300 bp paired-end sequences in six sequencing runs. Analyses were conducted in R (v.3.5.2)[72] using the Divisive Amplicon Denoising Algorithm (DADA) in the respective R package ('dada2', v.1.14.1)[73]. Demultiplexed paired-end reads were used as input for the dada2 analysis pipeline following a modified online tutorial[74]. In a first step, reads were trimmed right before the occurrence of the first ambiguous base (N). Subsequently, the forward and reverse primers were truncated from both reads using the cutadapt tool (v.1.9)[75]. Commonly, base qualities of the reads drop towards the 3′-end and need to be truncated accordingly. Forward reads were trimmed run-wise to a length between 250 and 280 bp, and reverse reads to a length of 220–260 bp. Trimming lengths were estimated based on visual inspection of the quality profiles specific to each sequencing run. In proportion to the trimming lengths, reads were filtered when their expected base errors, and sequence pairs exceeding run-depending values of 2.5–2.8 for the forward, and 2.2–2.6 for the reverse reads were removed. For each run-related sequence pool, error rates were learned, and sample inference and paired-end merging were performed according to the online tutorial. Subsequently, tables of amplicon sequence variants (ASVs) were compiled and merged into one, before predicting and removing chimeric sequences. Reads of length greater than 450 bp or shorter than 320 bp were removed from the dataset. Finally, taxonomy was assigned to each ASV based on a dada2-specific pre-prepared reference database from PR2 (v.4.12.0)[76], including eight different taxonomic ranks: Kingdom, Supergroup, Division, Class, Order, Family, Genus, and Species.

The relative proportion of 18S gene copy numbers is influenced by the relative abundance of the organisms in the sample, as well as by varying gene copy numbers[77]. However, it has been shown for various protist taxa that there is high correlation of 18S gene copy numbers to the cell volume, and that sequence abundance correlates with microscopic counts for larger taxa, which are easy to identify[78]. Thus, results of amplicon sequencing can provide a reliable tool to study eukaryotic microbial community compositions[79].

**Fatty acids.** To study the long-term diet signals in krill and salps, we extracted fatty acids from the tissue of the same individuals used for the stomach content analysis. Animals were lyophilized for 24 h at 1 bar (Zirbus, GOT 2000), and subsequently dry weight was measured. Before lipid extraction, whole lyophilized animals were homogenized, except for krill where the head and larger chitin components were removed. While krill specimens composed of one individual and had enough biomass for the analysis, two to three salp specimens were pooled per station, sex, and stage to increase biomass. The fatty acid and alcohol composition was analyzed based on a protocol by Kattner and Fricke[80]. Lipids were extracted with dichloromethane/methanol (2:1 v:v) from homogenized samples and subsequent transesterification was performed with 3% sulfuric acid in methanol for 4 h at 80 °C. The resulting fatty acid methyl esters (FAME) and fatty alcohols were extracted with cyclohexane. Subsequent analysis was performed on a gas chromatograph (Agilent 6890 N, Hewlett Packard) on a DB-FFAP column (30 m, 0.25 mm diameter, 0.25 μm film thickness), using a temperature program from 160–240 °C (4 °C min$^{-1}$, hold for 15 min). Where necessary, samples were diluted and analyzed for a second time on a 60 m column (same diameter and film thickness). The respective temperature program started at 80 °C, increased to 160 °C in steps of 20 °C min$^{-1}$, followed by an increase of 2 °C min$^{-1}$ to 240 °C, which was held for 20 min. Fatty acids and alcohols were identified based on known standards.

**Statistics and reproducibility**
*Sequencing data.* Amplicon sequencing of the 18S variable region V4 resulted in 15,227,483 raw reads in 292 samples, which were assigned to 9971 unique amplicon sequence variants (ASV). Technical krill replicates were pooled and the counts added together. Subsequently, predator DNA was removed from salp and krill samples respectively. ASVs, which were not assigned to a taxonomic level lower than Eukaryota (Phylum) were blasted (BLAST®, National Center for Biotechnology Information, Bethesda MD, USA) and removed in case no match with a higher identity than 98% was found. This resulted in a refined dataset containing 6,126,388 reads.

This refined dataset was analyzed following the notion that these data are compositional, analyzing ratios rather than absolute values[81]. For PCR and sequencing, the DNA volume is set to an artificial, unified concentration across all samples, thus total counts are not meaningful. ASVs with less than 100 counts overall and samples with less than 300 counts were removed and zero counts replaced applying a Bayesian-multiplicative replacement in the 'zCompositions'

package in R[82] (v.1.3.4). Subsequently, a centered-log-ratio (clr) transformation was conducted (i.e. log transformation of the geometric mean of the ratio transformed data).

Multivariate analyses were performed to evaluate the structure and variance of the data and to identify clusters using a principal component analysis (PCA) on the clr-transformed data. Differences among the five sampling groups (krill stomach content, salp stomach content, krill fecal pellets, salp fecal pellets, and plankton) were calculated using an ANOVA-like approach performing Welch's t-test and Wilcoxon rank sum test for all pairwise group comparisons based on 128 Monte Carlo replicates drawn from a Dirichlet distribution, taking the average significant value as representative using the 'ALDEx2' package in R[83] (v.1.19.4). In addition, we calculated the proportionality (equivalent to correlation in non-compositional data) between ASVs using the ρ-metric in the 'propr' package for R[84] (v.4.2.6) to find ASVs that had a constant or near-constant ratio variance across samples which were compositionally associated. Unsupervised clustering using Euclidean distances on the clr-transfomed data was performed to further investigate the observed clusters. All sequencing data analyses were conducted in R, v.3.6.1[72] partly modifying available scripts[85,86].

To assess the feeding selectivity of krill and salps, we calculated Ivlev's selectivity index using the 'dietr' package v.1.1.1 in R[87] based on the relative abundance of the clr-transfomed data on the taxonomic level of 'Class' for the plankton community (available prey) and the stomach content of krill and salps (diet). Ivlev's selectivity index ranges from +1 indicating preference of a prey taxon to –1 indicating avoidance, while 0 denotes random feeding[88].

*Fatty acid data.* Data of the identified fatty acids and fatty alcohols were expressed as percentages of total fatty acids and zero values were replaced by half the minimum positive value of the corresponding fatty acid per sample to be able to perform subsequent multivariate statistical analyses. We focused on the dietary marker fatty acids for three main groups: 16:1($n$−7) and 20:5($n$−3) for diatoms, 18:4($n$−3) and 22:6($n$−3) for dinoflagellates, and 20:1($n$−11/$n$−9/$n$−7), 22:1($n$−11/$n$−9/$n$−7) for calanoid copepods[89–91]. The proportional amounts of the single compunts of each of these marker FA were added up and are referred to as diatom, dinoflagellate, and copepod markers, respectively. The effect of the explanatory variables species (krill vs. salps) and region on fatty acid markers were tested using analysis of variance (ANOVA) after testing for the validity of assumptions (normal distribution, homogeneity of variance) in R, v.3.6.1[72]. In case the assumptions were not met, a non-parametric Kruskal-Wallis rank sum test was used and significant results inspected using the pairwise Wilcoxon rank sum test with Bonferroni adjustments. Additionally, we tested for the effect of sex and length for each species separately using the same approach. Multivariate analyses of fatty acid data were conducted using the easyCODA package (v.0.31.1) in R[92] using an unweighted log-ratio analysis.

**Reporting summary.** Further information on research design is available in the Nature Research Reporting Summary linked to this article.

## Data availability
Supplementary data, figures and tables are available in the supplementary information for this manuscript. The raw-, primer trimmed paired-end sequencing reads were deposited in the European Nucleotide Archive (ENA), project number PRJEB40056. Supplementary Movies 1 and 2 are available at figshare, https://doi.org/10.6084/m9.figshare.14216378.

## Code availability
Custom R code used for the analysis of the sequencing data is available from the GitHub repository https://github.com/ncpauli/Selectivity_SOGrazer.git.

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

## Acknowledgements

This study was part of the project "Population Shift and Ecosystem Response—Krill vs. Salps" (POSER) funded by the Lower Saxony ministry of science and culture (MWK) and the project "The performance of krill vs. salps to withstand in a warming Southern Ocean (PEKRIS) funded by the Federal Ministry of Education and Research (BMBF, 03F0746A), both headed by B.M. M.H.I and C.M.F. were supported by the HGF Young Investigator Group SeaPump "Seasonal and regional food web interactions with the biological pump": VH-NG-1000. C.M.F. was additionally supported by the AWI Strategy-Fund project EcoPump. We would like to thank the captain, crew, and chief scientist of RV Polarstern PS112 for their help and support during the cruise. A special thanks to Ryan Driscoll, Martina Vortkamp, and Larysa Pakhomova for the collection and measurements of krill and salps, and to Christin Konrad for his help with drift traps and video recordings. We would also like to thank Anna Friedrichs and Anne-Christin Schulz for their support to provide the water samples for further analyses. We are thankful to Valeria Adrian, Swantje Rogge, and Kerstin Oetjen for their help with laboratory work and to Sara Driscoll and Hannah Marchant for helpful comments on an earlier version of this manuscript.

## Author contributions

B.M., M.H.I., E.A.P. and N.-C.P. conceived the study. N.-C.P., M.H.I., T.H.B., E.A.P., C.M.F., and B.M. performed fieldwork. K.M. conceived the molecular work. Laboratory work was performed by N.-C.P.; S.N. and N.-C.P. performed the bioinformatic sequence analyses. M.G. coordinated fatty acid analyses. P.W. provided data on the vertical plankton distribution. All authors were involved in writing the manuscript and gave final approval for publication.

## Funding

## Competing interests

The authors declare no competing interests.
