## [Peer Review File · Communications Biology]

Reviewers' comments:

Reviewer #1 (Remarks to the Author):

General comments:

This is a fascinating and important study by a group of distinguished and experienced researchers in the field of Southern Ocean food web dynamics, with special expertise in salps and krill. The overall question or premise of the study is novel and of wide interest: that Southern Ocean salps and Antarctic krill share dietary preferences and that both are selective feeders. Despite my enthusiasm for the topic and my deep respect for the authors, the manuscript has a number of significant lacks that prevent my supporting its publication in the current form.

A key concern is that the authors refer repeatedly to "species diversity" of the diet of salps and krill based on DNA metabarcoding using a short region (V4) of the 18S rRNA "tree of life gene", which is evolutionarily conserved and usually considered to be a poor choice for detection and discrimination of species. Another key concern is lack of detail about collection and analysis of the organisms present in the "ambient" water column, which forms the basis of the claim for selective feeding of both species. Specific concerns and suggestions are detailed with reference to sections and line numbers of the manuscript.

Specific comments:

ABSTRACT:

The description of the study needs to include fundamental information about what DNA sequence markers (metabarcoding region(s)) were used and which fatty acids were analyzed. Without very general information about the technical approaches used, it is impossible to understand the results reported, or appreciate the implications and the comparisons to previous studies.

INTRODUCTION:

This section would benefit from reorganization. I suggest that previous studies of the diets of salps and krill be summarized in separate sections. The novelty of this paper is the integration and comparison of the impacts of the two species on the ecosystem, but previous studies have almost always focused on only one of these species. As written, the previous investigations of feeding of Antarctic salps and krill cannot be properly appreciated. Neither topic is sufficiently referenced, and in neither case is the evidence fully explained and carefully compared with consideration of diverse approaches used, including morphological / microscopic, ecological food web, isotopic, molecular metabarcoding, etc.

In addition, the broader implications of this paper might be more effectively introduced, summarized, and referenced as a foundation for the subsequent questions explored here. As written, the discussion is a curious juxtaposition of statements, with citations to both recent and long-ago studies, but does not provide a thoughtful or informative basis for the questions asked. Suggest reorganization into question-driven topics.

Line 56: Citation of Cox et al. (2018) #14 is curious, since that paper claims no krill decrease over the period reported.

Line 62: Why say salps are non-selective feeders here? If the authors wish to explain prior views which they seek to counter, that should be explained here. The topic of selective feeding – both in general and for the species of interest here - needs careful introduction, definition, and explanation. Curious that the citation for salps is Andersen (1998). There are more appropriate and much more recent papers that explore the issue of salp feeding.

Line 71: Wording could be improved: "will most likely trigger" is not appropriate when referring to speculation about the future in a scientific journal.

RESULTS:

Line 99: Restate to indicate sequencing of DNA extracted from stomach contents and fecal pellets.

Lines 113 and 134: The authors should address the issue of extremely high and variable 18S copy numbers in dinoflagellates, which may significantly impact – and invalidate – the results reported here. See Gong & Marchetti (2019) doi: 10.3389/fmars.2019.00219

Line 152-157: The RESULTS are summarized in terms of species diversity. There is no citation to any table showing these data, and all tables (including Supplementary Material) report results in terms of taxonomic groups. The DNA metabarcoding research community now generally agrees that 18S hypervariable regions (including the V4 region analyzed in this study) cannot reliably resolve and discriminate species, so reporting species diversity in the prey assemblage is not acceptable or justifiable. Most especially, the use of ASV (amplicon sequence variants from the DADA2 pipeline) as a proxy for species is not accurate or acceptable.

METHODS:

Lines 343-345: Additional details about the collection and analysis of “ambient” field samples is absolutely necessary. What mesh size did plankton net tows use? What filter size were CTD samples captured on? Without additional information about the water column sampling, it is impossible to evaluate the comparison with the krill and salp gut contents or to justify the conclusions of selective feeding of these species.

Line 449: Use of ASV numbers as a semi-quantitative measure of diet preference needs explanation and justification – and additional citations to similar studies using metabarcoding results of semi-quantitative (relative abundance or biomass) analyses.

Lines 460-461: The exclusion of non-Eukaryote DNA in the metabarcoding bioinformatics is curious, since salps may rely significantly on prokaryotic trophic resources. It would have been desirable to report, evaluate, and analyze these data before excluding them.

Reviewer #2 (Remarks to the Author):

Review of Pauli et al. “Selective feeding behavior of salps increases competition with krill – a paradigm shift?”

In this manuscript Pauli et al. present a large data set – including metabarcoding sequencing for krill and salp gut contents and fecal pellets, and for water column samples, as well as fatty acid results for krill and salps. Their study covers broad areas in the northern WAP region, with sampling in the autumn (March & April). They conclude from these data that both krill and salps exhibit selective feeding, because their gut contents and fecal pellets are different in composition to the water column samples. Based on this, they discuss potential competition for food resources between these species, and how this could affect the ways in which warming changes ecosystem dynamics in the Southern Ocean.

I found this to be an interesting manuscript, which was enjoyable to read and clearly contains some very interesting new data! The language is nearly flawless, and the authors are clearly very familiar with the relevant literature.

I was a bit surprised the authors did not include any selectivity analyses, when selective feeding seems to be their main conclusion, nor any calculations of niche overlap to address their point about increased competition.

I was also a bit concerned that there was no discussion of the potential limitations in the sampling scheme – specifically that a water filter from a single depth may not be representative of the prey field experienced by krill and salps collected over the top 200 meters of the water column.

I have some more specific suggestions below. Some of these are very small language suggestions – because the rest is so well written these small things stood out. Some of the suggestions are just ideas, and you are welcome to ignore them if you do not think they would add to your manuscript (I have marked these with asterisks, they do not require a reply)

Title: I understand the desire for a catchy title, but this seems a bit beyond the concrete findings of the manuscript. (unless I missed statistical findings on competition) Perhaps it could be toned down a little? "Selective feeding in both key Southern Ocean zooplankton species"? or "competitors or just neighbors – selective feeding in Antarctic krill and salps" or something along that vein.

Line 48: "its biomass" sounds wrong to my ear. Perhaps "their biomass" or "the species' biomass"?

Lines 52-56: This isn't as much a settled fact as the authors make out. Because of the difficulties in accurately surveying such a patchy organism in remote and broad areas, there is still contention over what the trends across time have been. I think the main point here is fine, but some mention of the contradictory results would add a useful nuance.

Line 60: Over what time interval did this distribution shift occur?

Line 67: Perhaps add "has" after "WAP"

Line 71: For an uncertain future I think "would" is more appropriate than "will" here

Line 73: "widely" does not work as an adjective with different. "wildly" does, but that is quite colloquial. Perhaps simply "very"?

Line 73: "Both" doesn't work here because grammatically it sets up a false comparison (i.e. both species differ from what?) "These two" might be a good replacement.

Lines 76-77: This sentence seems at odds with the rest of the paragraph. The rest of the paragraph seems to be discussing the ways krill and salps differ, but this sentence seems to be saying they both play a similar role in carbon export – so from that perspective changing the krill/salp balance wouldn't matter. If that's not what you were aiming for, perhaps something along the lines of: While both species play important additional roles in carbon cycling by generating sinking fecal pellets, these pellets are very different in morphology and density, such that changes in the balance between krill and salps can have implications for carbon export and the microbial loop.

Lines 103-105: It would be a nice addition here to include what each sample is, particularly since the methods section comes later. E.g. krill stomach contents (61 samples of 10 krill each), ambient plankton community (10 samples of 2L), etc.

Line 110: I was a little unsure what "all sampling groups" referred to here, if it's the five mentioned above, perhaps change to "all five sampling groups"?

Lines 152-157: Are these differences meaningful or significant? Is there a way you could estimate the confidence around them? E.g. leave-one-out analyses or rarefaction? If not, this information might be better suited to the supplementary section.

Lines 212-216: Some calculation of selectivity coefficients would be a logical addition here. There are certainly caveats because of differing digestion efficiency, but those also apply to the overall PCA analysis presented.

Line 225: Somewhere in this section it would be important to discuss the potential limitations of comparing zooplankton sampled across a broad depth range to a single net sample at one depth. Zooplankton can feed at different depths for a variety of reasons (patchy prey, predator avoidance, temperature preference, etc.), and phytoplankton/microzooplankton communities can vary strongly by depth, even over small vertical scales.

*Lines 236-240: Some diatom species are more heavily silicified than others. Are there different trends in diet vs gut contents when comparing different diatom species/genera? That could help show whether salps are breaking down some diatoms themselves (in which case less silicified diatoms would be more common in salp diet than in fecal pellets), or if these diatom signatures

come from krill fecal pellets.

*Lines 247-248: The point about flagellates colonizing fecal pellets is an interesting one! Can you test this by comparing across your fecal pellet samples? E.g. one might expect that samples from deeper traps would have more degraded "true" gut contents, but have had time to attract more flagellates? Do the aquarium generated fecal pellets have fewer flagellates than the trap collected fecal pellets?

Line 255: I'm not sure this is valid, given that you have just said the flagellates likely were not actually part of the diet. Perhaps phrase this a bit more cautiously.

Line 256: "dominated" doesn't seem appropriate when this fraction was only around a quarter.

Lines 286-287: What fraction of the sequences from salp fecal pellets are of salp? That would be important information to consider when assessing the possibility that these salp sequences in krill gut contents originate from feeding on salp fecal pellets.

Line 294: This paragraph is confusing because the topic sentence is only on dinoflagellates, but the paragraph goes on to also discuss crustaceans. Perhaps it could be rephrased?

Line 299: "as well as"

Line 300: This is unclear – if they are "unidentified" how were they "attributed" to krill? Are these sequences definitely from krill? Or is this an assumption because krill are the biomass most abundant malacostracan in the Southern Ocean?

Lines 303-309: Another possibility to consider is that syndiniales may have been infesting prey (such as copepods) of the krill/salps.

Line 323: Recruitment is traditionally used more as a point in time, e.g. after their first year krill recruit to the adult population, conditions over that year can lead to successes or failures in recruitment. Perhaps this could be rephrased as "longer pre-recruitment period", or "longer juvenile period" or similar?

Line 353: The time to catch on board is useful information (as that is the period of potential net feeding), but it is also important to know time to frozen (as the gut contents will be continuing to be digested over this entire period).

Lines 372-379: Sensors for which results are not presented at all in the manuscript could be eliminated from this section for simplicity.

Line 380: I did not see any information on no template controls. Presumably these were carried out at least at the PCR stage? Perhaps also at the DNA extraction and/or sequencing stages? Was any evidence of contamination observed?

Lines 406-408: This section was unclear to me. Pooling technical replicates is a good practice to minimize random biases which can appear in PCR, particularly with low template reactions. But I don't follow how this would increase the number of reads.

Lines 447-474: This section could very much benefit from more structure – paragraphs or subheadings or both.

Lines 448-452: I agree that illumina sequence data is compositional, but it is not for the reasons given here. When you prepared the libraries for sequencing, you presumably included whatever volume of purified PCR product would give the desired total DNA concentration – so at that stage one typically artificially sets all the samples to have roughly the same amount of DNA fragments (to ensure every sample gets enough data to analyze). That is why the total counts are not meaningful.

Lines 452-453 & 464: This clr transformation is discussed in two places. Was it performed twice? If not, could these be condensed?

Line 493: It would also be nice to include the R code used in the analyses.

Line 674: This is a useful and important table, but it is difficult to grasp the overarching sampling plan at a glance. It would probably be more helpful to readers to place a map of sampling locations here in the main text and shift this table over to the supplement. Quantarctica is an easy (and free) software for making nice looking station maps for Southern Ocean research.

*Figure 1: I think it is interesting that the salp fecal pellets are closer to the salp gut contents than the krill fecal pellets are to the krill gut contents? I guess this is a result either of the krill digesting things more efficiently or krill fecal pellets attracting more flagellates? It might be an interesting point to mention in the manuscript text.

Figure 2: I found the two shades of purple difficult to distinguish

Figure S3: This is a really interesting and important figure. Unfortunately, it is very difficult to read. There are way more categories listed than anyone could tell apart colors. Perhaps set some threshold, and pool everything less than X % into "other" (you could include an additional figure with just the rare groups if you want to present them). There needs to be labels of some kind on the x-axis.

Figure S6: The legends should not cover up the data points. It would be much easier to interpret if there was some logic to the color schemes. For example, stations in the same area could be similar colors. Is there any difference between the two panels on the left?

Reviewer #1 (Remarks to the Author)

General comments

This is a fascinating and important study by a group of distinguished and experienced researchers in the field of Southern Ocean food web dynamics, with special expertise in salps and krill. The overall question or premise of the study is novel and of wide interest: that Southern Ocean salps and Antarctic krill share dietary preferences and that both are selective feeders. Despite my enthusiasm for the topic and my deep respect for the authors, the manuscript has a number of significant lacks that prevent my supporting its publication in the current form.

Response: We appreciate the valuable comments and suggestions from the reviewer. We clarified and improved the manuscript according to the significant concerns that were raised by the reviewer and we think that this has significantly improved the manuscript.

A key concern is that the authors refer repeatedly to “species diversity” of the diet of salps and krill based on DNA metabarcoding using a short region (V4) of the 18S rRNA “tree of life gene”, which is evolutionarily conserved and usually considered to be a poor choice for detection and discrimination of species.

Response: We thank the reviewer for the remarks and take the concerns raised very seriously. We think that there is a misunderstanding about the way we have conducted the analyses of the 18S rRNA gene. The aim of this study was to investigate the diet and fecal pellet composition of krill and salps, which predominantly feed on plankton. Thus, we chose the 18S rRNA variable region V4 as marker region for our study. As shown by several studies, the 18S rRNA gene is a suitable tool to study microbial eukaryotic diversity, particularly in plankton communities (cf. Tragin et al. 2018 Environ. Microbiol, Lie et al. 2014 AEM, Metfies 2020 PLoS), while other barcodes, such as COI are more suitable for studies on metazoans (Bucklin et al. 2011 Ann. Rev. Mar. Sci.). We agree with the reviewer that the 18S rRNA gene does not allow for a discrimination on species level. This is why all analyses of the metabarcoding results in this study were conducted on the taxonomic level of ‘Class’ and only major plankton groups, such as diatoms and dinoflagellates, were compared.

We acknowledge that the term ‘species diversity’ was used at one single position in the discussion of the manuscript (Line 310) when referring to the results of the Shannon diversity analysis, when we actually referred to ASV diversity. We agree with the reviewer that the term ‘species’ in this context was a poor choice of words. We have removed the respective section to avoid any future misunderstandings

Another key concern is lack of detail about collection and analysis of the organisms present in the “ambient” water column, which forms the basis of the claim for selective feeding of both species. Specific concerns and suggestions are detailed with reference to sections and line numbers of the manuscript.

Response: We added details about the collection and analysis of the organisms of the ambient water column to the manuscript. Please see our response to comment 1.10 below for more details.

Specific comments reviewer #1

ABSTRACT:

Comment 1.1 The description of the study needs to include fundamental information about what DNA sequence markers (metabarcoding region(s)) were used and which fatty acids were analyzed. Without very general information about the technical approaches used, it is impossible to

understand the results reported, or appreciate the implications and the comparisons to previous studies.

Response: The abstract was edited to include information about the applied methods:

“Here, we provide a direct comparison of the diet and fecal pellet composition of krill and salps using 18S metabarcoding and fatty acid markers.” (Line 32-33).

We analyzed the entity of in total 30 single fatty acids and focused on three fatty acid combinations, which are established trophic markers for diatoms, dinoflagellates and copepods, respectively. Due to the length limitation of 150 words for the abstract, we have therefore refrained from mentioning the single analyzed fatty acids. We hope that the reviewer agrees with the changes shown above.

INTRODUCTION:

Comment 1.2 This section would benefit from reorganization. I suggest that previous studies of the diets of salps and krill be summarized in separate sections. The novelty of this paper is the integration and comparison of the impacts of the two species on the ecosystem, but previous studies have almost always focused on only one of these species. As written, the previous investigations of feeding of Antarctic salps and krill cannot be properly appreciated. Neither topic is sufficiently referenced, and in neither case is the evidence fully explained and carefully compared with consideration of diverse approaches used, including morphological / microscopic, ecological food web, isotopic, molecular metabarcoding, etc.

Response: The introduction was restructured and edited according to the reviewer’s comments and an additional paragraph on selective feeding was added:

“Both krill and salps are filter feeders, but differ in their feeding modes, potential prey-size spectrum and diet composition (Table 1). Krill are selective feeders with a diatom-dominated diet and were shown to prefer diatoms over smaller prymnesiophytes and cryptophytes in incubation experiments²²⁻²⁴. In contrast, the diet of salps mainly reflects the composition of the available plankton community, thus salps are assumed to be non-selective, indiscriminate feeders^{1,25}. Selective feeding, i.e. the selection of particular prey items while avoiding or rejecting others, is displayed by most zooplankton²⁶. The process of prey selection may be defined mechanically for example by the mesh size of the filtering apparatus (‘passive selection’), or can be based on chemical cues or mechano-receptors (‘active selection’), e.g. in krill²⁶⁻²⁸. In the pelagic environment with unevenly distributed food quantity and varying quality, prey selection is an important factor to balance the energetic costs of foraging against food quality and quantity and to adjust to changing conditions^{27,29}. Selective feeding mechanisms can exert strong control on nutrient turnover, primary production and biogeochemical processes²⁷. Consequently, the different feeding modes of krill and salps might have consequences for the plankton and grazer community structure at the WAP.” (Lines 71-84)

In addition, we have added an overview table summarizing previous studies on the feeding mode and diet composition of krill and salps (Table 1):

	Euphausia superba	Salpa thompsoni
Feeding mode	 • Feeding basket formed by thoracic legs, filtering of particles through fine net of setae^{40,89} • Feeding independent from active swimming, feeding rates can be adjusted^{24,40} 	 • Mucous net deployed in the pharyngeal cavity retaining particles from water pumped from the anterior to the posterior opening^{45,90} • Feeding and locomotion are continuous processes⁴⁵, no adjustment of feeding rates^{1,90}

Potential prey size-range	 • 2–3 μm to $\sim 1\text{ mm}$^{30,89} 	 • Submicron $<1\ \mu\text{m}$ to $> 1\text{ mm}$^{25,45,64}
Diet composition overview	 • Diatom-dominated diet, mainly herbivorous^{24,40} 	 • Food generalists, diet reflects available plankton community^{25,45}
Microscopy/visual inspection	 • Diatom dominated, autotrophic flagellates, dinoflagellates, tintinnids^{36,58} 	 • Diatom-dominated, radiolarians, silicoflagellates, dinoflagellates^{37,91}
Fatty acids	 • Diatoms, copepods, foraminifera, flagellates, athecate dinoflagellates^{30,58,928} 	 • Flagellates, moderate diatom contribution, copepods³⁷
Metabarcoding	 • Diatom-dominated, cercozoans and copepods³¹ 	 • Dinoflagellate-dominated, few diatoms³⁴
Other DNA-based methods	 • Diatom-dominated diet incl. silicoflagellates, copepods, cercozoa dinoflagellates, ciliates, cercozoans^{36,93} 	 • NA

Comment 1.3 In addition, the broader implications of this paper might be more effectively introduced, summarized, and referenced as a foundation for the subsequent questions explored here. As written, the discussion is a curious juxtaposition of statements, with citations to both recent and long-ago studies, but does not provide a thoughtful or informative basis for the questions asked. Suggest reorganization into question-driven topics.

Response: Thanks for pointing this out. We have reorganized and clarified both the introduction and the discussion. The introduction was revised to better introduce the topic of selective feeding and to introduce the research questions. Please see our response to comment 1.2 above for more details on the edits to the introduction.

Comment 1.4 Line 56: Citation of Cox et al. (2018) #14 is curious, since that paper claims no krill decrease over the period reported.

Response: The respective paragraph was rephrased, also in accordance with a comment by reviewer #2, emphasizing that a large-scale decline in krill abundances is debated in the scientific community: “From 1926 to 2016, a southward shift of krill and declining abundances north of 60°S have been observed¹⁰. However, it remains debated whether there is a large-scale decline in krill abundances in the SO^{10,15,18}.” (Lines 56-58)

Comment 1.5 Line 62: Why say salps are non-selective feeders here? If the authors wish to explain prior views which they seek to counter, that should be explained here. The topic of selective feeding – both in general and for the species of interest here - needs careful introduction, definition, and explanation. Curious that the citation for salps is Andersen (1998). There are more appropriate and much more recent papers that explore the issue of salp feeding.

Response: A paragraph to better introduce the topic of selective feeding of zooplankton in general, as well as for krill and salps in particular, was added (Lines 71-81, please see response to comment 1.2 above).

We agree that there are more recent studies on the feeding of salps, e.g. by von Harbou et al. 2011 reporting visual inspection and fatty acid results, as well as a recent molecular study by Metfies et al. 2014. Both studies are cited in this manuscript. In addition, we have added two recent papers on feeding mechanisms in doliolids and Mediterranean salp species to the introduction:

“Recent studies found indications for selective feeding of doliolids, a group closely related to salps, in the Atlantic Ocean³², and suggested that the diet of Mediterranean salps is determined by prey taxonomy, rather than size³³. Yet, there are no recent studies on the diet of salps in the Southern Ocean and studies that simultaneously compare the diet composition of krill and salps from the same region are lacking.” (Lines 90-94)

Most other recent studies focus on gut fluorescence methods and grazing rates (cf. Pakhomov 2006) as are thus not suitable to be referenced in the context of diet composition. We would also like to add that many detailed studies on feeding dynamics were provided by early workers in the field and many of these aspects have not been studied since, thus we consider it a necessity to also cite these early, original studies (cf. Alldredge & Madin 1982, Boyd 1984 and Suh & Nemoto 1987). If the reviewer has a particular reference in mind that we are not aware of, we are happy to include it in the manuscript.

Comment 1.6 Line 71: Wording could be improved: “will most likely trigger” is not appropriate when referring to speculation about the future in a scientific journal.

Response: The wording was changed to “...is expected to trigger a cascade ...”. (Line 65)

RESULTS:

Comment 1.7 Line 99: Restate to indicate sequencing of DNA extracted from stomach contents and fecal pellets.

Response: The sentence was changed as suggested:

“The diets of krill and salps were compared by sequencing genomic DNA extracted from the stomach contents and fecal pellets of organisms collected from the same regions along the AP.” (Lines 114-115).

Comment 1.8 Lines 113 and 134: The authors should address the issue of extremely high and variable 18S copy numbers in dinoflagellates, which may significantly impact – and invalidate – the results reported here. See Gong & Marchetti (2019) doi: 10.3389/fmars.2019.00219

Response: A paragraph discussing varying 18S copy numbers was added to the material and methods section:

“The relative proportion of 18S gene copy numbers is influenced by the relative abundance of the organisms in the sample, as well as by varying gene copy numbers⁷⁸. However, it has been shown for various protist taxa, that there is high correlation of 18S gene copy numbers to the cell volume, and that sequence abundance correlates with microscopic counts for larger taxa, which are easy to identify⁷⁹. Thus, results of amplicon sequencing can provide a reliable tool to study microbial community compositions⁸⁰.” (Lines 459-464).

Comment 1.9 Line 152-157: The RESULTS are summarized in terms of species diversity. There is no citation to any table showing these data, and all tables (including Supplementary Material) report results in terms of taxonomic groups. The DNA metabarcoding research community now generally agrees that 18S hypervariable regions (including the V4 region analyzed in this study) cannot reliably resolve and discriminate species, so reporting species diversity in the prey assemblage is not acceptable or justifiable. Most especially, the use of ASV (amplicon sequence variants from the DADA2 pipeline) as a proxy for species is not accurate or acceptable.

Response: We agree with the reviewer that in the context of the Shannon diversity the use of the term ‘species diversity’ was inappropriate and misleading. The respective paragraph was removed from the manuscript to avoid future misconceptions. We emphasize that all other analyses in the manuscript were conducted on the taxonomic level of “Class” for the particular reasons the reviewer has mentioned.

METHODS:

Comment 1.10 Lines 343-345: Additional details about the collection and analysis of “ambient” field samples is absolutely necessary. What mesh size did plankton net tows use? What filter size were CTD samples captured on? Without additional information about the water column sampling, it is impossible to evaluate the comparison with the krill and salp gut contents or to justify the conclusions of selective feeding of these species.

Response: We have clarified in the material and method section to show that no plankton net was used, but water samples for the analysis of the plankton community were taken from water sampled with a CTD (Lines 391-400). The information of the filters size is provided in the manuscript, line 402: “Water was filtered through a 0.4 μm membrane filter (Whatman Nucleopore, 47 mm polycarbonate) at 200 mbar...”.

Information on the mesh size of the plankton nets (IKMT; RMT) used to capture krill and salps for the stomach content analysis was added:

“For stomach content and tissue analyses, krill and salps were collected at ten stations across all regions by oblique net tows with an IKMT (Isaacs-Kidd Midwater Trawl, mesh size 505 μm), or with a RMT (Rectangular Midwater Trawl, mesh size 320 μm) in the upper 200 m of the water column.” (Lines 374-76).

In addition, a selectivity analysis was added, as well as data on the vertical distribution of the plankton community to further support the conclusion on selective feeding:

“Feeding selectivity

To assess the feeding selectivity of krill and salps, we calculated Ivlev’s selectivity index based on the relative abundance of taxa in the 18S libraries of the plankton community and the stomach content of krill and salps. The selectivity index ranges from +1 indicating preference of a prey taxon to –1 indicating avoidance, while 0 denotes random feeding³⁷. Across all stations, salps showed a strong preference for small flagellates (Filosa-Thecofilosea, +0.88; Figure 4), Syndiniales (+0.73), and Chrysophyceae (+0.78), while Pymnesiophytes (–0.90), Pelagophyceae (–0.87), Picozoa (–0.90), and Cryptophyceae (–0.96) were avoided. Krill selectively fed on copepods (+0.61), polychaetes (+0.58), and golden algae (Chrysophyceae, +0.89; Fig. 4), while avoiding radiolarian protozoans (Acantharia, –0.85) and Cryptophyceae (–0.71).

In addition to the plankton community sampled at a single depth in the chlorophyll maximum layer, at four of the ten stations water samples were collected at the surface, 100 m, and 200 m depth (Supplementary Figure 5). The preference of salps for Filosa-Thecofilosea was also observed across the different depths (>0.4; Supplementary Figure 6) and a preference for Syndiniales was observed at all stations except for one (St. 25). Similarly, the avoidance of pymnesiophytes and the other avoided groups was apparent across all depths (–0.47 to –1). For krill, the preference for copepods was apparent across all depths and the preference for polychaetes was observed at all stations except for one (St. 25).” (Lines 174–191)

Comment 1.11 Line 449: Use of ASV numbers as a semi-quantitative measure of diet preference needs explanation and justification – and additional citations to similar studies using metabarcoding results of semi-quantitative (relative abundance or biomass) analyses.

Response: Please see our response to comment 1.8 above

Comment 1.12 Lines 460-461: The exclusion of non-Eukaryote DNA in the metabarcoding bioinformatics is curious, since salps may rely significantly on prokaryotic trophic resources. It would have been desirable to report, evaluate, and analyze these data before excluding them.

Response: We fear that there is a misunderstanding. The sentence the reviewer is referring to states that sequences that were “...not identified further than Eukaryotes” were excluded. Thus, we excluded sequences, which could not be attributed to known sequences in the database. We blasted the respective sequences to ensure that no appropriate taxonomic assignment is possible. We added this information and rephrased the sentence to avoid any future misunderstandings:

“ASVs, which were not assigned to a taxonomic level lower than Eukaryota (Phylum) were blasted (BLAST®, National Center for Biotechnology Information, Bethesda MD, USA) and removed in case no match with a higher identity than 98% was found.” (Lines 486-491)

Reviewer #2 (Remarks to the Author)

Review of Pauli et al. “Selective feeding behavior of salps increases competition with krill – a paradigm shift?”

In this manuscript Pauli et al. present a large data set – including metabarcode sequencing for krill and salp gut contents and fecal pellets, and for water column samples, as well as fatty acid results for krill and salps. Their study covers broad areas in the northern WAP region, with sampling in the autumn (March & April). They conclude from these data that both krill and salps exhibit selective feeding, because their gut contents and fecal pellets are different in composition to the water column samples. Based on this, they discuss potential competition for food resources between these species, and how this could affect the ways in which warming changes ecosystem dynamics in the Southern Ocean.

I found this to be an interesting manuscript, which was enjoyable to read and clearly contains some very interesting new data! The language is nearly flawless, and the authors are clearly very familiar with the relevant literature. I was a bit surprised the authors did not include any selectivity analyses, when selective feeding seems to be their main conclusion, nor any calculations of niche overlap to address their point about increased competition. I was also a bit concerned that there was no discussion of the potential limitations in the sampling scheme – specifically that a water filter from a single depth may not be representative of the prey field experienced by krill and salps collected over the top 200 meters of the water column. I have some more specific suggestions below. Some of these are very small language suggestions –because the rest is so well written these small things stood out. Some of the suggestions are just ideas, and you are welcome to ignore them if you do not think they would add to your manuscript (I have marked these with asterisks, they do not require a reply)

Response: We appreciate the helpful and constructive comments to improve our manuscript. We added a selectivity analysis and were able to obtain additional data on the vertical plankton distribution to support our findings on selective feeding. We hope that the addition of these data have adequately addressed the issues highlighted by the reviewer and find that it has improved the manuscript. Please see more details in our responses to the specific comments below.

Specific comments

Comment 2.1 Title: I understand the desire for a catchy title, but this seems a bit beyond the concrete findings of the manuscript. (unless I missed statistical findings on competition) Perhaps it could be toned down a little? “Selective feeding in both key Southern Ocean zooplankton species”? or “competitors or just neighbors – selective feeding in Antarctic krill and salps” or something along that vein.

Response: We agree with the reviewer’s remark and changed the title to *“Selective feeding in Southern Ocean key grazers – diet composition of krill and salps.”*

Comment 2.2 Line 48: “its biomass” sounds wrong to my ear. Perhaps “their biomass” or “the species’ biomass”?

Response: Edited as suggested: *“... however over 50% of their biomass is located...”*. (Line 50)

Comment 2.3 Lines 52-56: This isn’t as much a settled fact as the authors make out. Because of the difficulties in accurately surveying such a patchy organism in remote and broad areas, there is still contention over what the trends across time have been. I think the main point here is fine, but some mention of the contradictory results would add a useful nuance.

Response: We agree with the reviewer that this should be phrased more carefully. The respective paragraph was changed to:

“From 1926 to 2016, a southward shift of krill and declining abundances north of 60°S have been observed¹⁰. However, it remains debated whether there is a large-scale decline in krill abundances in the SO^{10,15,18”}. (Lines 56-58).

Comment 2.4 Line 60: Over what time interval did this distribution shift occur?

Response: The respective time period was added to the sentence:

“Consequently, in response to climatic changes, an increase in the abundance of salps was observed and their southern distribution limit shifted from 60°S to 65°S since 1980^{1,15,16”}. (Lines 60–61)

Comment 2.5 Line 67: Perhaps add “has” after “WAP”

Response: Edited as suggested to: *“...the plankton community composition in the northern part of the WAP has shifted.”* (Line 88)

Comment 2.6 Line 71: For an uncertain future I think “would” is more appropriate than “will” here

Response: We agree with the reviewer’s remark. The wording was rephrased to:

“A long-term shift from krill to salps is expected to trigger a cascade of short- and long-term changes in the pelagic ecosystem of the western Atlantic sector of the SO.” (Line 65)

Comment 2.7 Line 73: “widely” does not work as an adjective with different. “wildly” does, but that is quite colloquial. Perhaps simply “very”?

Response: The wording was changed accordingly: *“...as the two organisms occupy very different ecological and spatial niches.”* (Line 66)

Comment 2.8 Line 73: “Both” doesn’t work here because grammatically it sets up a false comparison (i.e. both species differ from what?) “These two” might be a good replacement.

Response: The respective phrasing was edited as suggested: *“...Krill and salps differ remarkably in their life cycles..”*. (Line 67)

Comment 2.9 Lines 76-77: This sentence seems at odds with the rest of the paragraph. The rest of the paragraph seems to be discussing the ways krill and salps differ, but this sentence seems to be

saying they both play a similar role in carbon export – so from that perspective changing the krill/salp balance wouldn't matter. If that's not what you were aiming for, perhaps something along the lines of: While both species play important additional roles in carbon cycling by generating sinking fecal pellets, these pellets are very different in morphology and density, such that changes in the balance between krill and salps can have implications for carbon export and the microbial loop.

Response: As part of the restructuring of the introduction section according to the comments by reviewer #1, the respective paragraph was revised and the role of krill and salps in biogeochemical cycles is now mentioned at the beginning of the introduction:

“Krill and salps can re-package large amounts of the primary production into large, carbon-rich and fast settling fecal pellets, and thus play an important role in biogeochemical cycles and carbon export in the SO^{1,6-8}.” (Lines 47-49).

Comment 2.10 Lines 103-105: It would be a nice addition here to include what each sample is, particularly since the methods section comes later. E.g. krill stomach contents (61 samples of 10 krill each), ambient plankton community (10 samples of 2L), etc.

Response: Edited as suggested:

“To identify selective feeding behavior, we compared the stomach contents of krill (61 samples across 10 stations) and salps (60 samples across 10 stations) to the ambient plankton community (10 samples across 10 stations of 2 L each). In addition, the composition of fecal pellets (FP) produced by krill and salps were compared (n = 14 for krill FP, n = 11 for salp FP).” (Lines 115-119).

Comment 2.11 Line 110: I was a little unsure what “all sampling groups” referred to here, if it's the five mentioned above, perhaps change to “all five sampling groups”?

Response: Changed as suggested:

“A principal component analysis (PCA) of the refined dataset revealed three main clusters, which were associated with all five sampling groups (plankton community, stomachs, and fecal pellets; Figure 2).” (Line 123-124)

Comment 2.12 Lines 152-157: Are these differences meaningful or significant? Is there a way you could estimate the confidence around them? E.g. leave-one-out analyses or rarefaction? If not, this information might be better suited to the supplementary section.

Response: The respective paragraph was removed from the manuscript to avoid misunderstandings regarding the taxonomic level of the data analysis in response to a comment by reviewer #1.

Comment 2.13 Lines 212-216: Some calculation of selectivity coefficients would be a logical addition here. There are certainly caveats because of differing digestion efficiency, but those also apply to the overall PCA analysis presented.

Response: We agree with the reviewer's remark and have added a selectivity analysis based on the relative abundance of the plankton community from the single depth dataset. In addition, we were able to obtain data on the vertical distribution of the plankton community for four of the ten sampled stations from the surface to 200 m (see also comment below) and calculated the selectivity across the different depths to support our findings from the single depth analysis. A paragraph on the feeding selectivity of krill and salps was added to the results section. For convenience, the respective section is shown below:

“Feeding selectivity

To assess the feeding selectivity of krill and salps we calculated Ivlev's selectivity index based on the relative abundance of taxa in the 18S libraries of the plankton community and the stomach content of krill and salps. The selectivity index ranges from +1 indicating preference of a prey taxon to -1

indicating avoidance, while 0 denotes random feeding³⁷. Across all stations, salps showed a strong preference for small flagellates (*Filosa-Thecofilosea*, +0.88; Fig. 4), *Syndiniales* (+0.73), and *Chrysophyceae* (+0.78), while *Prymnesiophytes* (−0.90), *Pelagophyceae* (−0.87), *Picozoa* (−0.90), and *Cryptophyceae* (−0.96) were avoided. Krill selectively fed on copepods (+0.61), polychaetes (+0.58), and golden algae (*Chrysophyceae*, +0.89; Fig. 4), while avoiding radiolarian protozoans (*Acantharia*, −0.85) and *Cryptophyceae* (−0.71).

In addition to the plankton community sampled at a single depth in the chlorophyll maximum layer, at four of the ten stations water samples were collected at the surface, 100 m and 200 m depth (Suppl. Fig. S6). The preference of salps for *Filosa-Thecofilosea* was also observed across the different depths (>0.4; Suppl. Fig. S7) and a preference for *Syndiniales* was observed at all stations except for one (St. 25). Similarly, the avoidance of *prymnesiophytes* and the other avoided groups was apparent across all depths (−0.47 to −1). For krill, the preference for copepods was apparent across all depths and the preference for polychaetes was observed at all stations except for one (St. 25).” (Lines 174–191)

Comment 2.14 Line 225: Somewhere in this section it would be important to discuss the potential limitations of comparing zooplankton sampled across a broad depth range to a single Niskin bottle sample at one depth. Zooplankton can feed at different depths for a variety of reasons (patchy prey, predator avoidance, temperature preference, etc.), and phytoplankton/microzooplankton communities can vary strongly by depth, even over small vertical scales.

Response: We agree with the reviewer’s remark. We were able to obtain additional data on the vertical distribution of the plankton community and calculated a selectivity index across the different depths to support our previous conclusions on selective feeding, please see our response to the comment 2.13 above. In addition, we added a sentence on the feeding selectivity across different depths to the discussion section:

“However, the salp stomach content in this study was significantly different from the ambient plankton community, and salps selectively fed on *Filosa-Thecofilosea* (small flagellates) and golden algae over *prymnesiophytes* and other unicellular algae. The feeding selectivity of krill and salps was confirmed across different depths from the surface to 200 m at several stations.” (Lines 243–247).

Comment 2.15 *Lines 236-240: Some diatom species are more heavily silicified than others. Are there different trends in diet vs gut contents when comparing different diatom species/genera? That could help show whether salps are breaking down some diatoms themselves (in which case less silicified diatoms would be more common in salp diet than in fecal pellets), or if these diatom signatures come from krill fecal pellets.

Response: This is an interesting idea. Considering that the 18S is not a suitable tool to resolve communities at a high taxonomic resolution (e.g. species), while the silica content and frustules of diatoms may vary inter- and intraspecifically, depending on light and other environmental conditions, we think that our data do not provide a resolution good enough to reveal detailed insights into this topic. In the following, we provide a tentative evaluation of the diatom genera across the main sampling groups. However, this analysis was not included in the manuscript for the reasons mentioned above.

Comment 2.16 *Lines 247-248: The point about flagellates colonizing fecal pellets is an interesting one! Can you test this by comparing across your fecal pellet samples? E.g. one might expect that samples from deeper traps would have more degraded “true” gut contents, but have had time to attract more flagellates? Do the aquarium generated fecal pellets have fewer flagellates than the trap collected fecal pellets?

Response: This is an interesting idea that we investigated by testing for a trend in the relative abundance of ciliates (Spirotrichea), dinoflagellates (Dinophyceae) and other flagellates (Filosa-Thecofilosea) between fecal pellets collected from sediment traps at 100 and 300 m. The abundance of dinoflagellates and ciliates decreased with increasing depth, while for other flagellates there was no clear trend. We have edited the respective sentence in the manuscript, however due to the relatively low number of replicates for fecal pellets in comparison to stomach content samples, we have decided not to include the here presented analysis in the manuscript.

“Another potential reason for such high relative abundances in the excreted material might be that the flagellates colonized FP in the water column after they were produced. Dinoflagellates (Dinophyceae) and ciliates (Spirotrichea) showed a trend towards higher abundances in FP from 100 m compared to 300 m, indicating that those taxa graze on fecal pellets in the upper water column.” (Lines 281-285)

A comparison between the aquarium generated fecal pellets and the trap collected would be very interesting as well. However, we analyzed krill pellets from the sediment traps (see figure below), and salp pellets from the incubations on-board. Due to the fragile nature of salp fecal pellets, we were not able to collect salp pellets from the sediment traps. Thus, in this case a comparison would not be meaningful, as we would not be able to quantitatively separate the colonization mechanisms in the water column during settling of the pellets (sediment trap collected) from results we obtained from the aquarium incubations. We have also edited the sentence on the collection of fecal pellets in the M&M section to emphasize the different sampling schemes for krill and salp pellets:

“We collected FP produced by krill from free drifting sediment traps (hereafter drift traps), as well as freshly produced salp fecal pellets from incubation experiments.” (Lines 382–384)

Comment 2.17 Line 255: I'm not sure this is valid, given that you have just said the flagellates likely were not actually part of the diet. Perhaps phrase this a bit more cautiously.

Response: Edited as suggested:

"Here, salp FP contained a significantly higher share of diatoms than krill pellets (36.2% vs. 13.9%), suggesting that salp fecal pellets could sink faster than those produced by krill, coinciding with previous observations^{8,56}." (Lines 289-291).

Comment 2.18 Line 256: "dominated" doesn't seem appropriate when this fraction was only around a quarter.

Response: We agree with the reviewer and changed the phrasing accordingly:

"Crustaceans accounted for about one third of the prey sequences in krill stomachs, and were mainly represented by the copepod genera Calanus, Oithona, and Metridia, agreeing with previous stomach content studies on krill²⁴." (Lines 292-294)

Comment 2.19 Lines 286-287: What fraction of the sequences from salp fecal pellets are of salp? That would be important information to consider when assessing the possibility that these salp sequences in krill gut contents originate from feeding on salp fecal pellets.

Response: We agree with the reviewer that the proportion of salp sequences in salp fecal pellets is a good indicator for the potential feeding of krill on salp pellets and thank the reviewer for this remark. We have calculated the share of salp sequences in salp pellets and have added the following sentence to the respective paragraph in the discussion section:

"The presence of salp sequences in krill stomachs might also indicate krill feeding on salp FP, which is of particular interest as salp FP are thought to play an important role in the carbon cycle. However, salp sequences accounted for only a small share (2.1%) in salp FP, suggesting that the larger share of salp sequences in the diet of krill derives from feeding on salps or salp remains." (Lines 321-325)

Comment 2.20 Line 294: This paragraph is confusing because the topic sentence is only on dinoflagellates, but the paragraph goes on to also discuss crustaceans. Perhaps it could be rephrased?

Response: The paragraph was shifted to merge with another paragraph discussing the diet composition of salps and was rephrased:

“In this study, the diet of salps was dominated by (dino-) flagellates, which is in accordance with previous results from the Lazarev Sea³⁴. Moreover, fatty acid analyses have indicated that salps have a flagellate-based diet year-round with a moderate amount of diatoms, and the overall fatty acid composition of salps agrees with previous studies in the WAP region^{36,45}. Besides flagellates, crustaceans accounted for about 13% of the diet of salps, including copepods, as well as unidentified Malacostraca sequences, which were blasted and identified to be krill sequences.” (Lines 257-262).

Comment 2.21 Line 299: “as well as”

Response: Added as suggested. (Line 261)

Comment 2.22 Line 300: This is unclear – if they are “unidentified” how were they “attributed” to krill? Are these sequences definitely from krill? Or is this an assumption because krill are the biomass most abundant malacostracan in the Southern Ocean?

Response: The sentence was edited for more clarity:

“Besides flagellates, crustaceans accounted for about 13% of the diet of salps, including copepods, as well as unidentified Malacostraca sequences, which were blasted and identified to be krill sequences.” (Lines 260-262)

Comment 2.23 Lines 303-309: Another possibility to consider is that syndiniales may have been infesting prey (such as copepods) of the krill/salps.

Response: We agree with the reviewer. We have edited the sentence accordingly with an additional reference: *“Another possible reason for the presence of Syndiniales in stomach and FP samples could be the ingestion of infected prey, such as copepods or other dinoflagellates⁶⁷.”* (Lines 332–334)

Comment 2.24 Line 323: Recruitment is traditionally used more as a point in time, e.g. after their first year krill recruit to the adult population, conditions over that year can lead to successes or failures in recruitment. Perhaps this could be rephrased as “longer pre-recruitment period”, or “longer juvenile period” or similar?

Response: Changed as suggested to *“...while krill depend on a longer juvenile development.”* (Line 348)

Comment 2.25 Line 353: The time to catch on board is useful information (as that is the period of potential net feeding), but it is also important to know time to frozen (as the gut contents will be continuing to be digested over this entire period).

Response: The time until the animals were shock frozen was added to the respective sentence:

“The catch was on-board and inside within five minutes and the animals were measured, sexed, staged, frozen in liquid nitrogen, and stored individually at –80 °C within approximately ten minutes.” (Line 376-377).

Comment 2.26 Lines 372-379: Sensors for which results are not presented at all in the manuscript could be eliminated from this section for simplicity.

Response: Removed as suggested.

Comment 2.27 Line 380: I did not see any information on no template controls. Presumably these were carried out at least at the PCR stage? Perhaps also at the DNA extraction and/or sequencing stages? Was any evidence of contamination observed?

Response: Information of the template controls included in this study was added to the respective section:

“Two to three template controls without DNA were carried through all steps of the PCR to account for potential contamination. In addition, template controls were randomly included in the library preparation steps. We did not find evidence for contamination.” (Lines 426-428).

Comment 2.28 Lines 406-408: This section was unclear to me. Pooling technical replicates is a good practice to minimize random biases which can appear in PCR, particularly with low template reactions. But I don't follow how this would increase the number of reads.

Response: The section was rephrased for more clarity:

“Amplification of DNA isolated from krill stomach samples with the blocking probe resulted in a lower total DNA yield. Therefore, we amplified and sequenced the stomach content samples of krill in triplicates to be able to pool the sequencing information from different reactions, as well as to account for the variability between samples caused by the expected low sequencing yield.” (Lines 435-438)

Comment 2.29 Lines 447-474: This section could very much benefit from more structure – paragraphs or subheadings or both.

Response: The “statistical analyses” section was restructured, condensed, and subheadings and more paragraphs were added. The respective section of the revised manuscript is as follows:

*“Statistical analyses and reproducibility
Sequencing data*

The initial amplicon sequencing of the 18S variable region V4 resulted in 15,227,483 raw reads in 292 samples, which were assigned to 9,971 unique amplicon sequence variants (ASV). Technical krill replicates were pooled and the counts added together. Subsequently, predator DNA was removed from salp and krill samples respectively. ASVs, which were not assigned to a taxonomic level lower than Eukaryota (Phylum) were blasted (BLAST®, National Center for Biotechnology Information, Bethesda MD, USA) and removed in case no match with a higher identity than 98% was found. This resulted in a re-fined dataset containing 6,126,388 reads.

This refined dataset was analyzed following the notion that these data are compositional, analyzing ratios rather than absolute values⁷⁸. For PCR and sequencing, the DNA volume is set to an artificial, unified concentration across all samples, thus total counts are not meaningful. ASVs with less than 100 counts overall and samples with less than 300 counts were removed and zero counts replaced applying a Bayesian-multiplicative replacement in the ‘zCompositions’ package in R⁷⁹ (v.1.3.4). Subsequently, a centered-log ratio (clr) transformation was conducted (i.e. log transformation of the geometric mean of the ratio transformed data).

Multivariate analyses were performed to evaluate the structure and variance of the data and to identify clusters using a principal component analysis (PCA) on the clr-transformed data. Differences among the five sampling groups (krill stomach content, salp stomach content, krill fecal pellets, salp fecal pellets, and plankton) were calculated using an ANOVA-like approach performing Welch’s t-test and Wilcoxon rank sum test for all pairwise group comparisons based on 128 Monte Carlo replicates drawn from a Dirichlet distribution, taking the average significant value as representative using the ‘ALDEx2’ package in R⁸⁰ (v.1.19.4). In addition, we calculated the proportionality (equivalent to correlation in non-compositional data) between ASVs using the p-metric in the ‘propr’ package for R⁸¹ (v.4.2.6) to find ASVs that had a constant or near-constant ratio variance across samples which were compositionally associated. Unsupervised clustering using Euclidean distances on the clr-transformed data was performed to further investigate the observed clusters. The prey selectivity of krill and salps was calculated using Ivlev’s selectivity index in the ‘dietr’

package v.1.1.1 in R⁸² using the relative abundance of the clr-transformed data on the taxonomic level of 'Class' for the plankton community (available prey) and the stomach content of krill and salps (diet). All sequencing data analyses were conducted in R, v.3.6.1⁷² partly modifying available scripts^{83,84}.

Fatty acid data

Data of the identified fatty acids and fatty alcohols were expressed as percentages of total fatty acids and zero values were replaced by half the minimum positive value of the corresponding fatty acid per sample to be able to perform subsequent multivariate statistical analyses. We focused on the dietary marker fatty acids for three main groups: 16:1(n-7) and 20:5(n-3) for diatoms, 18:4(n-3) and 22:6(n-3) for dinoflagellates, and 20:1(n-11/n-9/n-7), 22:1(n-11/n-9/n-7) for calanoid copepods⁸⁵⁻⁸⁷. The proportional amounts of each of these marker FA were added up and are referred to as diatom, dinoflagellate, and copepod markers, respectively. The effect of the explanatory variables species and region on fatty acid markers were tested using analysis of variance (ANOVA) after testing for the validity of assumptions (normal distribution, homogeneity of variance) in R, v.3.6.1⁷². In case the assumptions were not met, a non-parametric Kruskal-Wallis rank sum test was used and significant results inspected using the pairwise Wilcoxon rank sum test with Bonferroni adjustments. Additionally, we tested for the effect of sex and length for each species separately using the same approach. Multivariate analyses of fatty acid data were conducted using the easyCODA package (v.0.31.1) in R⁸⁸ using an unweighted log-ratio analysis." (Lines 483-529).

Comment 2.30 Lines 448-452: I agree that illumina sequence data is compositional, but it is not for the reasons given here. When you prepared the libraries for sequencing, you presumably included whatever volume of purified PCR product would give the desired total DNA concentration – so at that stage one typically artificially sets all the samples to have roughly the same amount of DNA fragments (to ensure every sample gets enough data to analyze). That is why the total counts are not meaningful.

Response: We agree that this part was not phrased well. The respective sentence was edited to: "This refined dataset was analyzed following the notion that these data are compositional, analyzing ratios rather than absolute values⁷⁸. For PCR and sequencing, the DNA volume is set to an artificial, unified concentration across all samples, thus total counts are not meaningful." (Lines 492-494).

Comment 2.31 Lines 452-453 & 464: This clr transformation is discussed in two places. Was it performed twice? If not, could these be condensed?

Response: We thank the reviewer for this remark and agree that this section benefits from being condensed. The clr-transformation was only performed once and the respective section was edited accordingly. Please see the section copy pasted as response to comment 2.29 above.

Comment 2.32 Line 493: It would also be nice to include the R code used in the analyses.

Response: An R Script is now available via the GitHub repository (https://github.com/ncpauli/Selectivity_SOGrazer.git). A code availability statement was added to the manuscript.

Comment 2.33 Line 674: This is a useful and important table, but it is difficult to grasp the overarching sampling plan at a glance. It would probably be more helpful to readers to place a map of sampling locations here in the main text and shift this table over to the supplement. Quantartica is an easy (and free) software for making nice looking station maps for Southern Ocean research.

Response: We agree with the reviewer that an overview of the sampling scheme is easier to grasp for the reader and we thank the reviewer for bringing Quantartica to our attention. We added a map

with the different stations and sampling devices using the proposed software (Figure 1). The original table was moved to the supplementary material file as suggested (Supplementary Table 2).

Figure 1:

Comment 2.34 *Figure 1: I think it is interesting that the salp fecal pellets are closer to the salp gut contents than the krill fecal pellets are to the krill gut contents? I guess this is a result either of the krill digesting things more efficiently or krill fecal pellets attracting more flagellates? It might be an interesting point to mention in the manuscript text.

Response: We agree with the reviewer and thank for the suggestion. A sentence was added to the discussion section:

“Overall, the composition of salp FP was more similar to their stomach content than krill FP to krill’s diet, suggesting that krill more efficiently digest their prey than salps.” (Lines 275–276)

Comment 2.35 Figure 2: I found the two shades of purple difficult to distinguish

Response: The colors were adjusted accordingly.

Figure 3:

Comment 2.36 Figure S3: This is a really interesting and important figure. Unfortunately, it is very difficult to read. There are way more categories listed than anyone could tell apart colors. Perhaps set some threshold, and pool everything less than X % into “other” (you could include an additional figure with just the rare groups if you want to present them). There needs to be labels of some kind on the x-axis.

Response: The figure was adjusted according to the reviewers comment. Rare taxa were grouped in a separate category (“others”) and sample IDs were added as x-axis labels.

Supplementary Figure 1:

Figure S6: The legends should not cover up the data points. It would be much easier to interpret if there was some logic to the color schemes. For example, stations in the same area could be similar colors. Is there any difference between the two panels on the left?

Response: The figure was edited accordingly and we adjusted legends and color schemes. The original figure showed the single stations, as well as the stations per area on the left side. For a better overview, we now only show the PCA with data points per area, while the colors were adjusted to highlight adjacent areas.

Supplementary Figure 4:

REVIEWERS' COMMENTS:

Reviewer #1 (Remarks to the Author):

The authors have clearly described their responses to all my comments on the earlier version of their paper. In nearly all cases, the revisions are sufficient to address the issue or question posed. In some cases (in my opinion), the revisions do represent significant improvements in the manuscript. One example of a major improvement is the addition of text (Lines 71-81) and Table 1, which now provide a detailed and thoroughly referenced summary of previous studies of Southern Ocean salp and krill diets. The reorganization of the Introduction and Discussion have successfully addressed nearly all the concerns I expressed in my review of the initial submission. Very nicely done!

I would like to call attention to a few remaining issues:

The addition of explicit definitions and quantitative analysis of prey selectivity, including the new Table 4, are excellent. The summary of RESULTS (Lines 174-191) includes detail that may be more appropriate for the METHODS section. This might be used to enhance the current description of prey selectivity in METHODS (Lines 509-512), perhaps creating a separate clearly-written paragraph or section devoted to this topic.

Lines 459-464: The revised text describes the issue of highly variable 18S copy numbers in protists, but ends with a summary statement about "microbial community compositions", with a citation about protists (#80). Protists are not properly considered microbes, so this needs to be re-written.

Overall, the manuscript is very much improved and I recommend publication following minor revisions.

REVIEWERS' COMMENTS:

Reviewer #1 (Remarks to the Author):

The authors have clearly described their responses to all my comments on the earlier version of their paper. In nearly all cases, the revisions are sufficient to address the issue or question posed. In some cases (in my opinion), the revisions do represent significant improvements in the manuscript. One example of a major improvement is the addition of text (Lines 71-81) and Table 1, which now provide a detailed and thoroughly referenced summary of previous studies of Southern Ocean salp and krill diets. The reorganization of the Introduction and Discussion have successfully addressed nearly all the concerns I expressed in my review of the initial submission. Very nicely done!

We would like to thank the reviewer once again for the helpful comments that have greatly improved our manuscript.

I would like to call attention to a few remaining issues:

The addition of explicit definitions and quantitative analysis of prey selectivity, including the new Table 4, are excellent. The summary of RESULTS (Lines 174-191) includes detail that may be more appropriate for the METHODS section. This might be used to enhance the current description of prey selectivity in METHODS (Lines 509-512), perhaps creating a separate clearly-written paragraph or section devoted to this topic.

Response: We agree with the reviewers' remark. We have moved the first two sentences of the original paragraph (Lines 175-178) to the method section, where we have edited and separated the paragraph on the selectivity analysis for more clarity:

"To assess the feeding selectivity of krill and salps, we calculated Ivlev's selectivity index using the 'diatr' package v.1.1.1 in R⁸⁷ based on the relative abundance of the clr-transformed data on the taxonomic level of 'Class' for the plankton community (available prey) and the stomach content of krill and salps (diet). Ivlev's selectivity index ranges from +1 indicating preference of a prey taxon to -1 indicating avoidance, while 0 denotes random feeding⁸⁸." (Lines 509-513)

Lines 459-464: The revised text describes the issue of highly variable 18S copy numbers in protists, but ends with a summary statement about "microbial community compositions", with a citation about protists (#80). Protists are not properly considered microbes, so this needs to be re-written.

Response: We have rephrased the sentence according to the reviewers comment. It now reads as follows: "Thus, results of amplicon sequencing can provide a reliable tool to study eukaryotic microbial communities community compositions⁷⁹." (Line 461)

Overall, the manuscript is very much improved and I recommend publication following minor revisions.